
# Dust transport and horizontal fluxes measurement with spaceborne lidars ALADIN, CALIOP and model reanalysis data

Guangyao Dai[1], Kangwen Sun[1], Xiaoye Wang[1], Songhua Wu[1,2,3], Xiangying E[1], Qi Liu[1], Bingyi Liu[1,2]

[1] Department of Marine Technology, College of Information Science and Engineering, Ocean University of China, Qingdao, 266100, China
[2] Laboratory for Regional Oceanography and Numerical Modelling, Pilot National Laboratory for Marine Science and Technology (Qingdao), Qingdao, 266200, China
[3] Institute for Advanced Ocean Study, Ocean University of China, Qingdao, 266100, China

*Correspondence to*: Songhua Wu (wush@ouc.edu.cn)

**Abstract.** In this paper, a long-term large-scale Sahara dust transport event occurred during 14 June and 27 June 2020 is tracked with the spaceborne lidars ALADIN and CALIOP observations and the models ECMWF and HYSPLIT analysis. We evaluate the performance of the ALADIN and CALIOP on the observations of dust optical properties and wind fields and explore the capability in tracking the dust events and in calculating the dust horizontal mass fluxes with the combination of measurement data from ALADIN and CALIOP coupled with the products from ECMWF and HYSPLIT. Compared with the traditional assessments based on the data from CALIOP and models, the complement of Aeolus-produced aerosol optical properties and wind data will significantly improve the accuracy of dust horizontal flux estimations. The dust plumes are identified with AIRS/Aqua Dust Score Index and with the Vertical Feature Mask products from CALIPSO. The emission, dispersion, transport and deposition of the dust event are monitored using the data from HYSPLIT, CALIPSO and AIRS/Aqua. With the quasi-synchronization observations by ALADIN and CALIOP, combining the wind vectors and relative humidity, the dust horizontal fluxes are calculated. From this study, it is found that the dust event generated on 14 and 15 June 2020 from Sahara Desert in North Africa, and then dispersed and transported westward over the Atlantic Ocean, and finally deposited in the Atlantic Ocean, the Americas and the Caribbean Sea. During the transport and deposition processes, the dust plumes are trapped in the Northeasterly Trade-wind zone between the latitudes of 5 °N and 30 °N and altitudes of 0 km and 6 km (in this paper we name this space area as "Saharan dust eastward transport tunnel"). From the measurement results on 19 June 2020, influenced by the hygroscopic effect and mixing with other types aerosols, the backscatter coefficients of dust plumes are increasing along the transport routes, with $3.88\times10^{-6} \pm 2.59\times10^{-6}$ m$^{-1}$sr$^{-1}$ in "dust portion during emission phase", $7.09\times10^{-6} \pm 3.34\times10^{-6}$ m$^{-1}$sr$^{-1}$ in "dust portion during development phase" and $7.76\times10^{-6} \pm 3.74\times10^{-6}$ m$^{-1}$sr$^{-1}$ in "dust portion during deposition phase". Finally, the horizontal fluxes at different dust parts and heights on 19 June and on entire transport routine during transportation are computed. On 19 June, the dust horizontal fluxes are about $2.17\pm1.83$ mg$\cdot$m$^{-2}\cdot$s$^{-1}$ in dust portion during emission phase, $2.72\pm1.89$ mg$\cdot$m$^{-2}\cdot$s$^{-1}$ in dust portion during development phase and $3.01\pm2.77$ mg$\cdot$m$^{-2}\cdot$s$^{-1}$ in dust portion during deposition phase. In the whole life-time of the dust event, the dust horizontal





fluxes are about $1.30\pm1.07$ mg$\cdot$m$^{-2}\cdot$s$^{-1}$ on 15 June 2020, $2.62\pm1.88$ mg$\cdot$m$^{-2}\cdot$s$^{-1}$ on 16 June 2020, $2.72\pm1.89$ mg$\cdot$m$^{-2}\cdot$s$^{-1}$ on 19 June 2020, $1.98\pm1.41$ mg$\cdot$m$^{-2}\cdot$s$^{-1}$ on 24 June 2020 and $2.11\pm1.74$ mg$\cdot$m$^{-2}\cdot$s$^{-1}$ on 27 June 2020. From this study, it is found that the minimum of the fluxes appears when the dust event is initially generated on 15 June. During the dust development stage, the horizontal fluxes gradually increase and reach to the maximum value on 19 June with the enhancement of the dust event. Then, the horizontal fluxes gradually decrease since most of the dust deposited in the Atlantic Ocean, the Americas and the Caribbean Sea. Combining the Chlorophyll concentrations data provided by MODIS-Aqua, the Saharan Dust is found transported across the oligotrophic regions Atlantic Ocean towards the Americas and Caribbean Sea, which are also oligotrophic regions. The mineral dust delivers micronutrients including soluble Fe and P to the deposition zones and has the potential to fertilizing the ocean and increase the primary productivity in the Atlantic Ocean and Caribbean Sea.

## 1 Introduction

The global aerosol distribution and wind profiles have significant impacts on the atmospheric circulation, marine–atmosphere circulation and aerosol activities. As the most abundant aerosol types in the global atmosphere, the mineral dust influences the radiation budget, air quality, climate and weather via direct and various indirect radiative effects. Mineral dust is also considered as a major source of nutrients for ocean and terrestrial ecosystems. By the prevailing wind systems, mineral dust deposited over the ocean and land surface and hence significantly affect the carbon cycle and perturb the ocean and land geochemistry (Velasco-Merino et al., 2018; Banerjee et al., 2019). The atmospheric mineral dust can be transport over tens of thousands of kilometers away from its source regions (Uno et al., 2009, Haarig et al., 2017, Hofer et al., 2017). For instant, the biggest dust source Africa produced over half the global total dust (Huneeus et al., 2011) and African dust transports westward over the Atlantic Ocean and reaches South America (Yu et al., 2015; Prospero et al., 2020), the Caribbean Sea (Prospero and Lamb, 2003) and southern United States (Bozlaker et al., 2013). Hence, the continuous observations of the dust long-range transport are crucial. As one of the best techniques for remotely studying the characteristics and properties of aerosols, lidar contributes much to measure the dust activities. As introduced in previous papers, several comprehensive field campaigns including Aerosol Characterization Experiment ACE-Asia (Huebert et al., 2003; Shimizu et al., 2004), the Puerto Rico Dust Experiment PRIDE (Colarco et al., 2003; Reid et al., 2003), the Saharan Dust Experiment SHADE (Tanré et al., 2003), the Saharan Mineral Dust Experiments SAMUM-1 (Heintzenberg, 2009) and SAMUM-2 (Ansmann et al., 2011), the Dust and Biomass-burning Experiment DABEX (Haywood et al., 2008), the Dust Outflow and Deposition to the Ocean project DODO (McConnell et al., 2008), the Pacific Dust Experiment PACDEX (Huang et al., 2008), the China-US joint dust field experiment (Huang et al., 2010), the Saharan Aerosol Long-Range Transport and Aerosol-Cloud-Interaction Experiment SALTRACE (Weinzierl et al., 2017), the study of Saharan Dust Over West Africa SHADOW, and the Central Asian Dust Experiment CADEX (Hofer et al., 2017, 2020a, 2020b) were conducted.

However, the measurement data from these campaigns are still not able to meet the requirements for the investigation of global dust impact on climate, ocean/land geochemistry and ecosystems. Therefore, the spaceborne lidars that are capable of





observing aerosol have become effective instruments and are widely used in terms of dust plume measurements. The satellite-
based lidar CALIOP (Cloud-Aerosol Lidar with Orthogonal Polarization) carried by the platform of CALIPSO (Cloud-Aerosol
Lidar and Infrared Pathfinder Satellite Observations) provides us the backscatter coefficient and extinction coefficient at the
wavelengths of 532 nm and 1064 nm (Winker et al., 2009). Additionally, the CALIOP product Vertical Feature Mask product
(VFM) presents the aerosol sub-types classification and the global dust events could be marked. Moreover, large efforts still
should be undertaken to monitor the dust emission, transport, dispersion, deposition and to explore dust impact on the Earth's
radiation, climate and ecosystem. Hence, the vertical profiling of the global wind field is necessary to calculate the dust fluxes.
Thanks to the efforts of the European Space Agency (ESA), a first ever spaceborne direct detection wind lidar, Aeolus, which
is capable of providing the globally high spatial and temporal vertical wind profiles is developed under the framework of
Atmospheric Dynamics Mission (ADM) (Stoffelen et al., 2005; ESA, 1999; Reitebuch et al., 2012; Kanitz et al., 2019). On 22
August 2018, the Aeolus was successfully launched onto its sun-synchronous orbit at a height of 320 km (Witschas et al., 2020;
Lux et al., 2020). A quasi-global coverage is achieved daily (~ 15 orbits per day) and the orbit repeat cycle is 7 days (111
orbits). The orbit is sun-synchronous with a local equatorial crossing-time of ~ 6 am/pm. The Atmospheric Laser Doppler
Instrument (ALADIN) is a direct detection high spectral resolution wind lidar carried by Aeolus and provides the vertical
profiles of the Line-of-Sight (LOS) wind speeds. It is operating at the wavelength of 354.8 nm. In order to retrieve the LOS
wind speeds, the Doppler shifts of light caused by the emotion of molecules and aerosol particles need to be identified. Aiming
at this, a Fizeau interferometer is applied in the Mie channel to extract the frequency shift of the narrow-band particulate return
signal by means of fringe imaging technique (Mckay, 2002). In the Rayleigh channel, two coupled Fabry-Perot interferometers
are used to analyze the frequency shift of the broad-band molecular return signal by the double edge technique (Chanin et al.,
1989; Flesia and Korb, 1999).

In the simultaneous observations of the dust plume, the aerosol optical properties can be obtained by means of ALADIN
and CALIOP. By further using the wind field data from ALADIN, the wind field and relative humidity data from ECMWF
and the trajectories from HYSPLIT model, the dust transport route and dust fluxes can be calculated. The paper is organized
as follows: in Section 2 the satellite-based instruments, ECMWF and HYSPLIT models are introduced. Section 3 presents the
details to the joint dust measurement strategy and methodology. In section 4 we provide the results and discussions on the dust
transport measurements on 19 June 2020 and during the whole lifetime of the dust event.

**2. Spaceborne instruments and meteorological models**

**2.1 ALADIN/Aeolus**

ALADIN, which is the unique payload of Aeolus, is a direct detection high spectral resolution wind lidar. It is a pulsed
ultraviolet lidar works at the wavelength of 354.8 nm with a laser pulse energy around 65 mJ and with a repetition of 50.5 Hz.
As the receiver, a 1.5 m diameter telescope is equipped for the collection of the backscatter light. The high spectral resolution
design of ALADIN allows for the simultaneous detection of the molecular (Rayleigh) and particle (Mie) backscattered signals




in two separate channels, each sampling the wind in 24 vertical height bins with a vertical range resolution between 0.25 km and 2.0 km. This makes it possible to deliver winds both in clear and (partly) cloudy conditions down to optically thick clouds at the same time. The horizontal resolution of the wind observations is about 90 km for the Rayleigh channel and about 10 km for the Mie channel. The detailed descriptions of the instrument design and a demonstration of the measurement concept are

introduced in e.g. Reitebuch et al., 2009, 2012; Straume et al., 2018; ESA 2008; Marksteiner et al., 2013; Kanitz et al., 2019; Witschas et al., 2020 and Lux et al., 2020.

The data products of Aeolus are processing at different levels mainly including Level 0 (instrument housekeeping data), Level 1B (engineering-corrected HLOS winds), Level 2A (aerosol and cloud layer optical properties), Level 2B (meteorologically-representative HLOS winds) and Level 2C (Aeolus-assisted wind vectors) (Tan et al., 2008, 2017). Within

the Level 2B processor, the Rayleigh-clear and Mie-cloudy winds are classified and the temperature and pressure correction are applied for the Rayleigh wind retrieval (Witschas et al., 2020). In this study, the Level 2A aerosol optical properties and Level 2C wind vectors are used. For the calculation of particle volume concentration distribution and mass concentration, the extinction coefficients and backscatter coefficients at the wavelength of 355 nm are used.

## 2.2 CALIOP/CALIPSO

Launched in 2006, CALIPSO is providing us the valuable aerosol and cloud optical properties information, e.g., particle depolarization ratio, extinction coefficient, backscatter coefficient and Vertical Feature Mask (VFM) (Winker et al., 2009). The VFM product describes the vertical and horizontal distribution of cloud and aerosol types along the observation tracks of CALIPSO. In this study, the backscatter coefficients and extinction coefficients at the wavelengths of 532 nm and 1064 nm are used for the calculation of the dust volume concentration distribution and mass concentration. The VFM from CALIPSO

are also applied to identify the subtypes of aerosol layers.

## 2.3 ECMWF climate reanalysis

Supported by the Copernicus Climate Change Service (C3S), ECMWF is providing the atmospheric reanalysis ERA5 which presents a detailed record of the global atmosphere, land surface and ocean waves from 1950 onwards (Hersbach et al., 2020). The 4D-Var assimilated ERA5 is producing the hourly vertical profiles (at 37 pressure levels) of global wind fields with a grid

resolution of 31 km. After the successful launch of the Aeolus, the ECMWF is starting to simulate the wind products of Aeolus from January of 2020. In this study, the wind field data from ECMWF is applied in filling in the missing data within the region between the tracks Aeolus of and CALIPSO and in connecting the data from these two spaceborne lidars.

## 2.4 HYSPLIT

The Hybrid Single-Particle Lagrangian Integrated Trajectory model (HYSPLIT) is a modelling system for determine the

trajectories, the transport and dispersion of air masses developed by The National Oceanic and Atmospheric Administration (NOAA) Air Resources Laboratory (ARL) (Draxler and Hess 1998; Draxler and Rolph 2012). It is widely used in studying





the development of atmosphere. The backward trajectory and forward trajectory are the mostly used model applications to determine the origin of air masses (Stein et al., 2015). In this study, the HYSPLIT is used in describing and re-checking the routes of the dust plumes transport, dispersion, and deposition.

**3. Methodology**

In the study of dust transport and fluxes measurement, as shown in Figure 1, the dust identification, Aeolus and CALIPSO tracks match, data analysis and the HYSPLIT model analysis are necessary and the schematic flowchart is described briefly. To identify the dust events and to choose the quasi-synchronization observations with ALADIN and CALIOP, the flowchart is presented in this figure. To preliminarily determine the occurrences of dust events, the "Dust score index" data provided by

AIRS/Aqua are used to determine the dust plume coverage and transport route. With this given information, the VFM products from the simultaneously observations with the spaceborne lidar CALIOP are applied to cross-check the identification of dust events. Hence the vertical distributions of dust plumes are presented. To find the original sources and to predict the transport routes of dust plumes, the backward trajectory and forward trajectory is used respectively. When the dust events are determined, the simultaneous observations with ALADIN and CALIOP have to be selected. Starting from the CALIOP observations, the

nearest Aeolus footprints could be figured out. Since the orbits of Aeolus and CALIPSO are different, they cannot meet each other at the exactly same time and same location. From our study, the CALIPSO is about 4 hours ahead of Aeolus. Base on the transport directions of dust events modelled with HYSPLIT, the tracks of Aeolus should be always downwind of the tracks of CALIPSO. When the tracks of Aeolus and CALIPSO are selected, the distances between the tracks can be calculated. If the distances are less than 200 km, the following procedures could be continued. In this study, the backscatter coefficient and

extinction coefficient at 355 nm from ALADIN, at 532 nm and 1064 nm from CALIOP are collected as the useful dataset. The backscatter coefficients and extinction coefficients at 355 nm correspond to the "Aeolus Level 2A Product" retrieved by SCA (standard correct algorithm). In this study, we choose SCA instead of ICA (iterative correct algorithm) because the extinction coefficients from ICA are noisy and the assumption of "one single particle layer filling the entire range bin" in SCA is reasonable and is met in the situation of the heavy dust events observation. The backscatter coefficients and extinction

coefficients at 532 nm and 1064 nm are the "Total_Backscatter_Coefficient_532", "Extinction_Coefficient_532", "Backscatter_Coefficient_1064" and "Extinction_Coefficient_1064". Since the footprints of Aeolus and CALIPSO are not exactly matched, the missing wind data between their tracks have to be filled in using the ERA5 wind field data.

In Figure 2, the flowchart of dust fluxes calculation procedure is provided. Based on the dataset consists of the backscatter coefficients and extinction coefficients at the wavelengths of 1064 nm and 532 nm from CALIOP and those at the wavelength

of 355 nm from ALADIN, the aerosol volume concentration distribution can be calculated based on regularization method which was performed by generalized cross-validation (GCV) from Müller et al. (1999). The advantage of this method is that it does not require prior knowledge of the shape of the particle size distribution and the measurement uncertainty is on the order of 20%. After integrating and multiplying an assuming typical dust particle density which is set as 2.65 $\mathrm{g \cdot cm^{-3}}$ referring





to previous studies (e.g., Schepanski et al., 2009; Hofer et al., 2017; Mamouri and Ansmann, 2017), the particle mass
concentration would be estimated as Engelmann et al. (2008) introduced. Combining with the horizontal wind speed provided
by Aeolus and ECMWF, when the relative humidity is lower than 90%, the aerosol mass fluxes can be computed with eddy
covariance (EC) method by Eq. (1).

$$Flux_{\text{aerosol-mass}} = \overline{m' \cdot v'}, \tag{1}$$

where $m$ is the aerosol mass concentration and $v$ is the horizontal wind speed.








**Figure 1. Dust identification, Aeolus and CALIPSO tracks match and data procedures.**

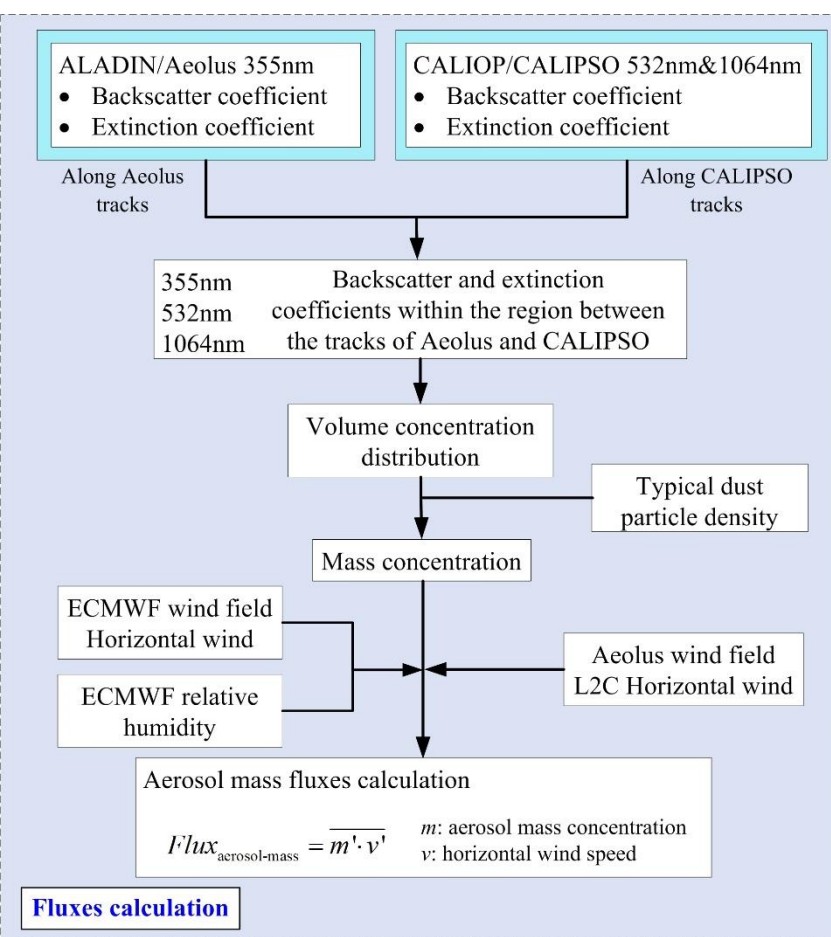

**Figure 2. The flowchart of the dust fluxes calculation procedure.**

## 4. Results and discussion

### 4.1 Measurement case with CALIOP, ALADIN, ECMWF and HYSPLIT

During 14 and 27 June 2020, a complete dust event process, including dust emission, transportation, dispersion and deposition, took place in the regions of Africa, Atlantic Ocean and the Americas. In Figure 3, the Dust Score Index provided by AIRS/Aqua at different stages are presented. From this figure, the long-term dust event generated on 14 and 15 June 2020 from Sahara Desert in North Africa, and then dispersed and transported westward over the Atlantic Ocean, and finally deposited in the west part of Atlantic Ocean, the Americas and the Caribbean Sea. It should be emphasized that since the dust scores are provided per day, the dust events are just preliminarily classified. From the spaceborne CALIOP observations, it is found that sometimes





dust events are actually present but are misjudged by the AIRS/Aqua, which may result from the interference from the high-altitude suspended cloud layers.

To cross-check the transport route of the dust events, as shown in Figure 4, the backward trajectories and forward trajectories
starting at 0500UTC 20 June 2020 with NOAA HYSPLIT model are conducted. In these trajectories, the heights of 2 km, 3 km and 4 km are chosen as the level heights.

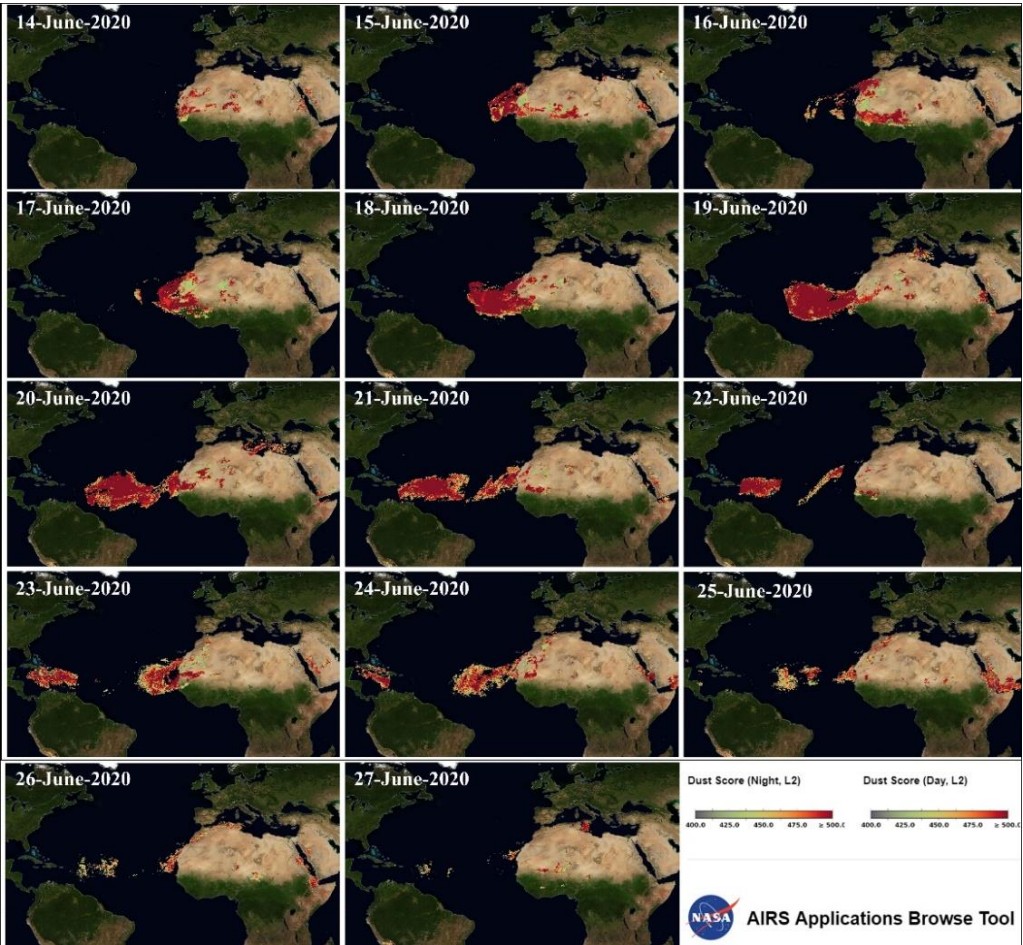

**Figure 3. The Dust Score Index provided by AIRS/Aqua at different stages, including emission, transportation, dispersion and deposition.**

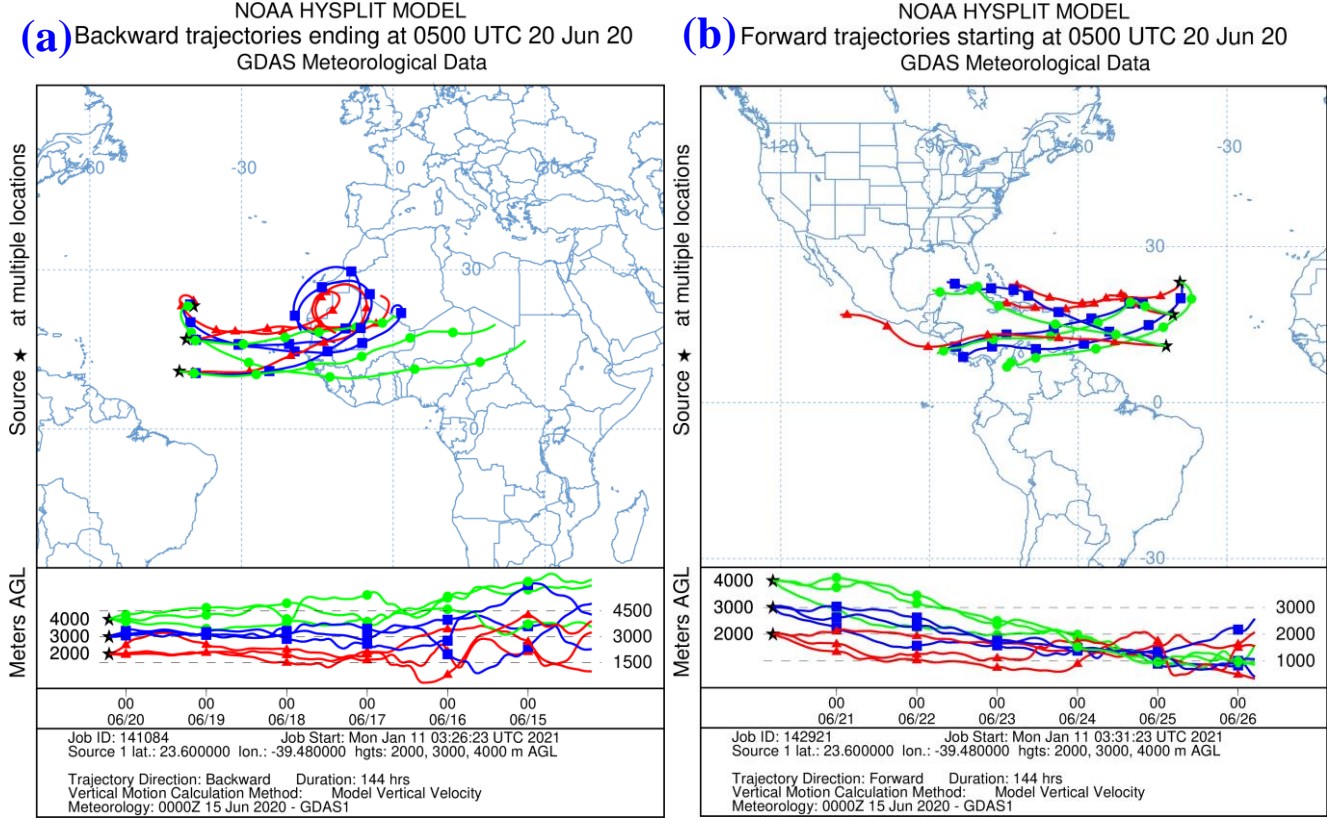

**Figure 4. (a) HYSPLIT backward trajectories ending at different sites and different heights on 20 June 2020; (b) HYSPLIT forward trajectories starting at different sites and different heights on 20 June 2020. The trajectories' durations are 144 hours.**

From Figure 4 (a), it is found that the dust plumes at about 4 km are generated from the central of Sahara Desert while the dust plumes at about 3 km and 2 km are occurred from the west side of the Sahara Desert. In Figure 4 (b), it is clearly indicated that the dust plumes were separated into two directions toward the Caribbean Sea and the Central/South America respectively. And an obvious deposition process of dust plumes is observed. After 26 June, transported over the whole Atlantic Ocean, most of the dust plumes were settled in the western Atlantic Ocean, the Central America and the Caribbean Sea.





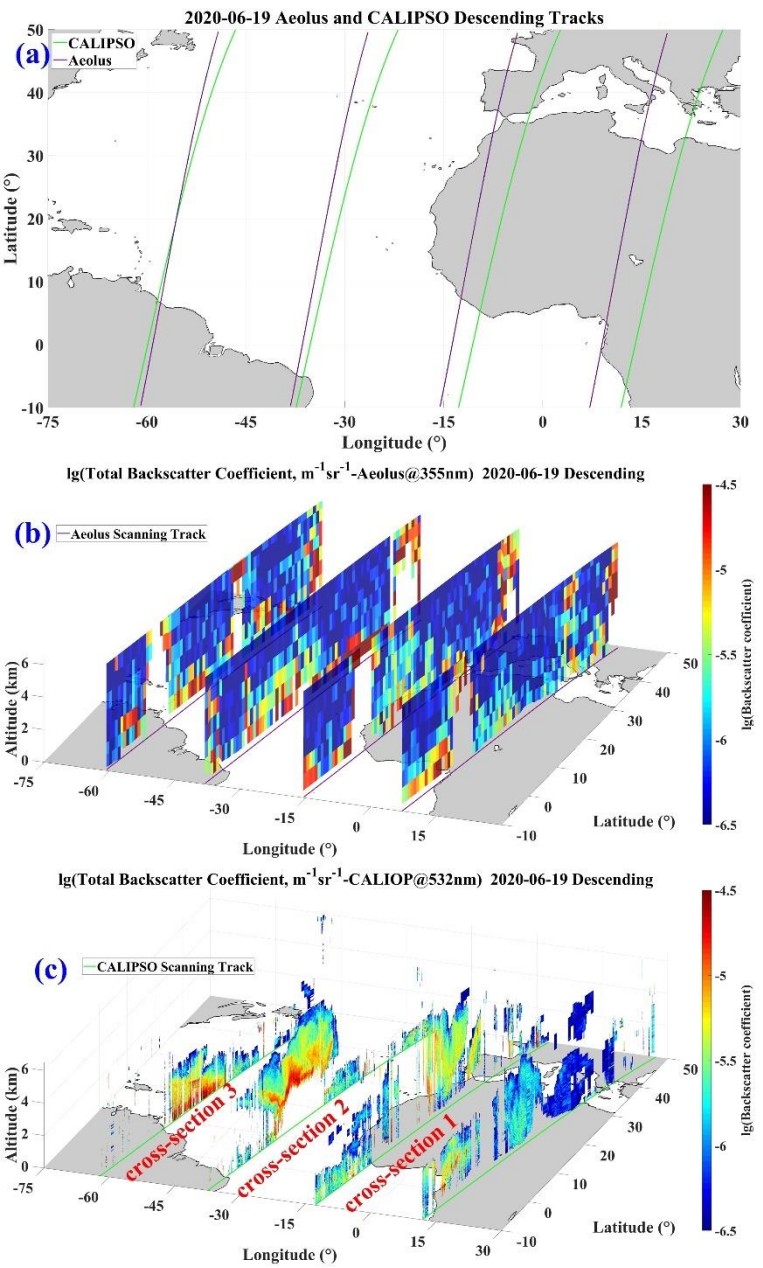

**Figure 5. Observation tracks of Aeolus and CALIPSO on 19 June 2020. The purple lines indicate the tracks of Aeolus and the green lines indicate the tracks of CALIPSO. (a) Vertical view of Aeolus and CALIOP tracks and HYSPLIT trajectories; (b) Backscatter coefficient cross-sections measured with Aeolus and (c) Total backscatter coefficient cross-sections measured with CALIOP.**

In this section, the dust event measurement case occurred on 19 June 2020 is introduced in detail. As shown in Figure 5, the quasi-synchronization observations with ALADIN and CALIOP and the total available backscatter coefficient profiles at three wavelengths measured on 19 June 2020 are presented, where the purple lines indicate the tracks of Aeolus and the green lines



indicate the tracks of CALIPSO. From the profiling of dust optical properties, discriminated by the CALIOP measurements, the dust dispersion over Atlantic Ocean on this day could be figured out. The extinction coefficients and backscatter coefficients at the wavelengths of 355 nm, 532 nm and 1064 nm within the dust mass are also determined. From the profiling,

it can be found that the mean backscatter coefficients at 532 nm are about $3.88\times10^{-6}\pm2.59\times10^{-6}$ m$^{-1}$sr$^{-1}$ in "cross-section 1",

$7.09\times10^{-6}\pm3.34\times10^{-6}$ m$^{-1}$sr$^{-1}$ in "cross-section 2" and $7.76\times10^{-6}\pm3.74\times10^{-6}$ m$^{-1}$sr$^{-1}$ in "cross-section 3". Since the dust plumes in "cross-section 1" are suspended over continental while the dust plumes in "cross-section 2" are suspended over ocean, probably influenced by the hygroscopic effect, the backscatter coefficients in "cross-section 2" are larger than that of "cross-section 1". However, the dust layers in "cross-section 3" are connected to the surface of ocean and mixed well with

other aerosol types (e.g., marine aerosol), hence the backscatter coefficients increase significantly.

    Based on the backscatter coefficients and extinction coefficients at 355nm, 532nm and 1064nm, combining the wind field data from ALADIN and ECMWF, the column dust transport fluxes can be calculated. The L2C wind product provided by Aeolus are result from the background assimilation of the Aeolus HLOS winds in the ECMWF operational prediction model. The u and v components of the wind vector and supplementary geophysical parameters are contained in L2C data product.

From the literature report (e.g., Lux et al., 2020), the Aeolus L2B Rayleigh LOS winds and the ECMWF model LOS winds are compared and the result shows good agreement with a correlation coefficient of 0.92 and mean bias of 1.62 ms$^{-1}$. Hence, in this study, both the ECMWF wind vector data and the "analysis_zonal_wind_velocity" and "analysis_meridional_wind_velocity" from Aeolus could be applied for the calculation of the dust fluxes.

    To calculated the dust fluxes during this event, the wind field and relative humidity information are necessary. Since the

observations with ALADIN and CALIOP are not exactly simultaneous, the stability of wind field between the scanning tracks of them has to be estimated. Hence, the wind speed, wind direction and relative humidity between the tracks are analysed with the data from ECMWF, as presented in Figure 6. From this figure, the wind speed, wind direction and relative humidity at the height surfaces of 1 km, 2 km, 3 km, 4 km, 5 km and 6 km are shown as examples. The wind fields between the tracks of Aeolus and CALIPSO at all height surfaces are smoothly distributed and the values are stable. Thus, the mean values of speed

and directions are applied in the calculation of dust fluxes. It should be emphasized that, during the calculations of the dust fluxes, the results with relative humidity higher than 90% have to be removed.

    In Figure 7, the dust horizontal fluxes at different heights of the three cross-sections are presented. It can be found that, dominated by the Northeasterly Trade-wind between the latitudes of 5 °N and 30 °N , the dust plumes are mainly transported to the west part of the Atlantic Ocean. From the profiling, it is figured out that the dust horizontal fluxes are about

$2.17\pm1.83$ mg$\cdot$m$^{-2}\cdot$s$^{-1}$ in "cross-section 1" (dust portion during emission phase), $2.72\pm1.89$ mg$\cdot$m$^{-2}\cdot$s$^{-1}$ in "cross-section 2" (dust portion during development phase) and $3.01\pm2.77$ mg$\cdot$m$^{-2}\cdot$s$^{-1}$ in "cross-section 3" (dust portion during deposition phase). The peak values of the dust horizontal fluxes appear in the "cross-section 3", which may result from the higher dust mass concentration in this section.


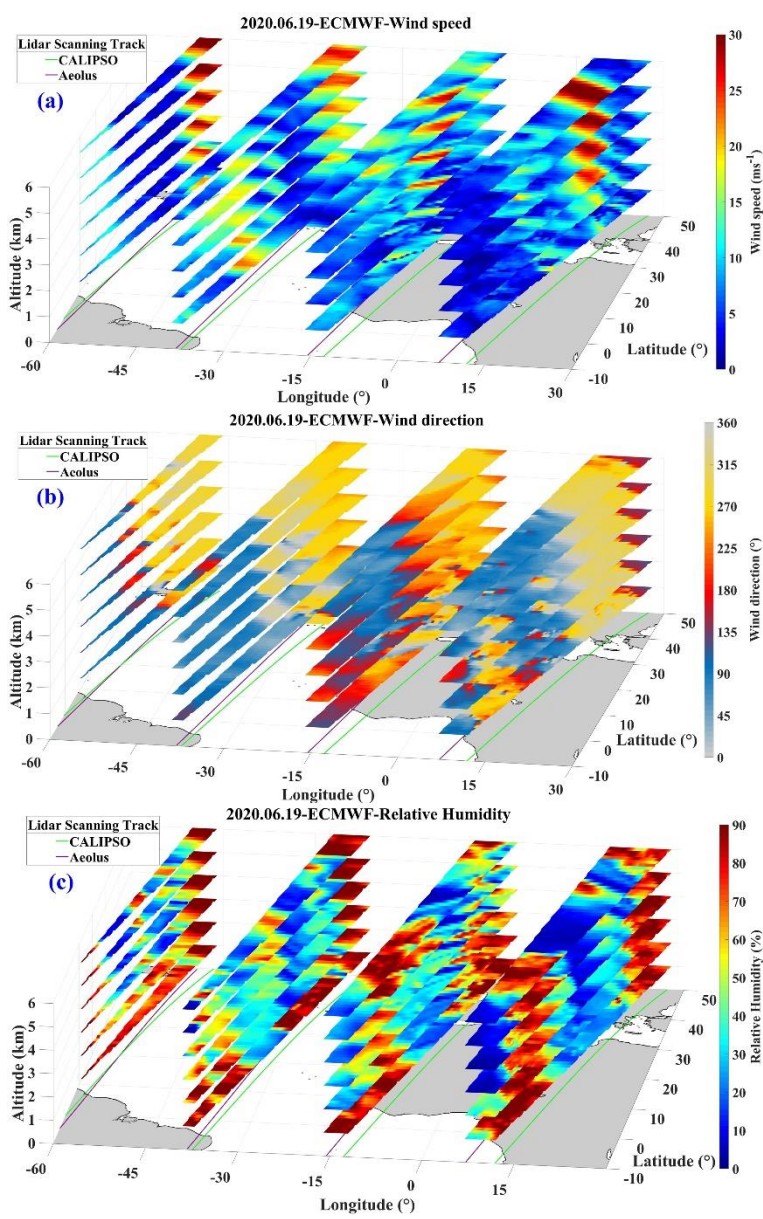

**Figure 6. The wind speed, wind directions and relative humidity between the quasi-synchronization observation tracks of Aeolus and CALIPSO provided by ECMWF on 19 June 2020.**





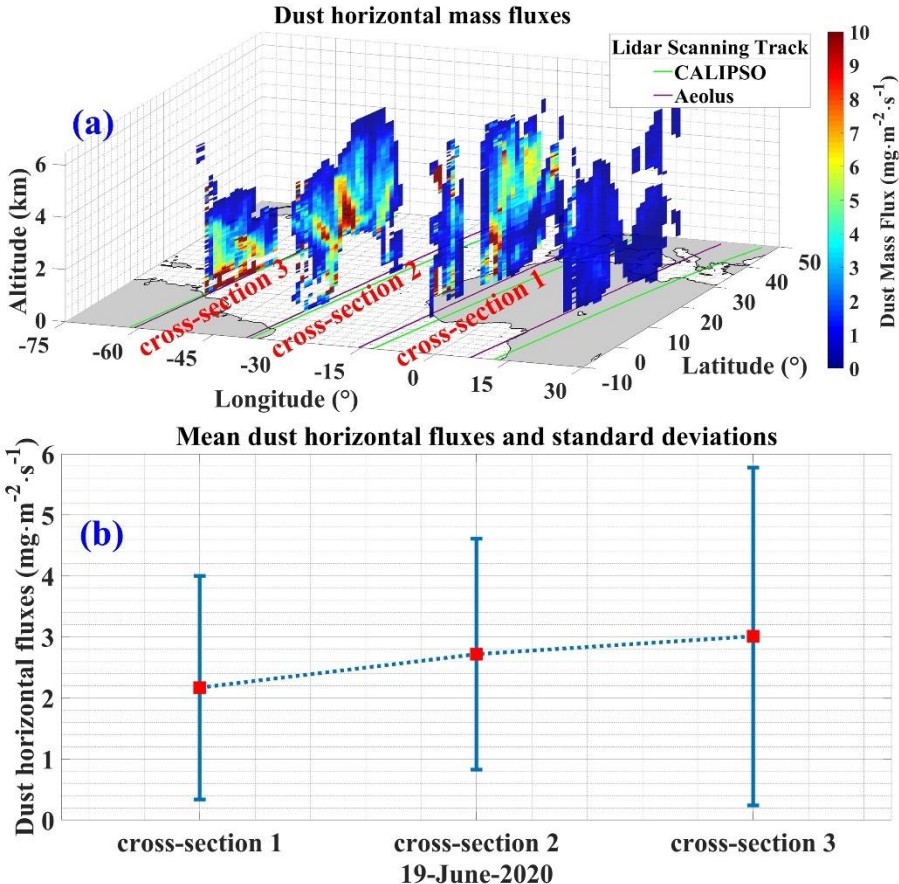

**Figure 7. The dust horizontal fluxes calculated with data from ALADIN, CALIOP and ECMWF (a) at different cross-sections of dust plumes and (b) the corresponding mean dust fluxes and standard deviations on 19 June 2020.**

### 4.2 Horizontal fluxes during the lifetime of dust event during 14 June and 27 June 2020

During this dust event, the quasi-synchronization observations with ALADIN and CALIOP are selected to follow the transport and dispersion of dust. As shown in Figure 8, the detailed information about the ALADIN and the CALIOP observations on 15, 16, 19, 24, 27 June 2020 along the transport route and the HYSPLIT modelling are shown. In Figure 8(a), the spaceborne lidars ALADIN and CALIOP quasi-synchronization scanning tracks on those days are indicated by dark purple lines and green lines, respectively. Additionally, the forward trajectories starting from 19 June and backward trajectories ending at 19 June are modelled and presented in dark red lines and light purple lines, respectively. In Figure 8(b) and (c), 5 cross-sections of backscatter coefficient at 355nm measured at different times with Aeolus and 5 cross-sections of backscatter coefficient at 532nm measured at different times with CALIOP are plotted, respectively. From these figures, we can find that the dust transport modelled with HYSPLIT match well with the dust masses at different cross-sections of Aeolus and CALIPSO. In Figure 8(d), the heights of the dust masses during the transportation at different days are modelled as well with HYSPLIT.





Consistent with the observations from ALADIN and CALIOP in Figure 8(b) and (c), there is an apparent deposition process along the transport route of the dust event.

In Figure 9, the wind speed and directions at certain height surfaces between the tracks of CALIPSO and Aeolus are shown and are smoothly distributed and the values are stable. The relative humidity is presented as well.





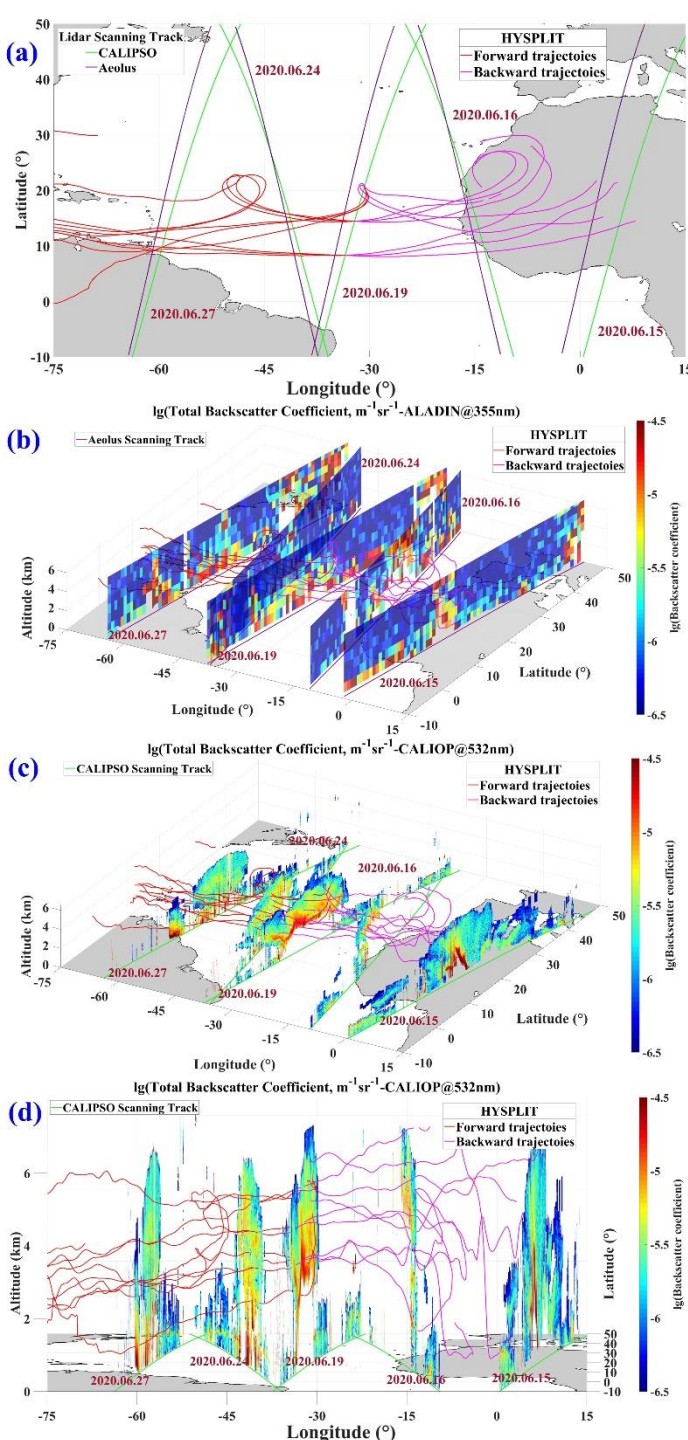


**Figure 8. Observation of dust event during 15 and 27 June 2020 with ALADIN and CALIOP. (a) Vertical view of Aeolus and CALIPSO tracks and HYSPLIT trajectories; (b) Backscatter coefficient cross-sections measured with ALADIN and HYSPLIT trajectories; (c) Total backscatter coefficient cross-sections measured with CALIOP and HYSPLIT trajectories and (d) Side view of HYSPLIT trajectories and cross-sections measured with CALIOP.**





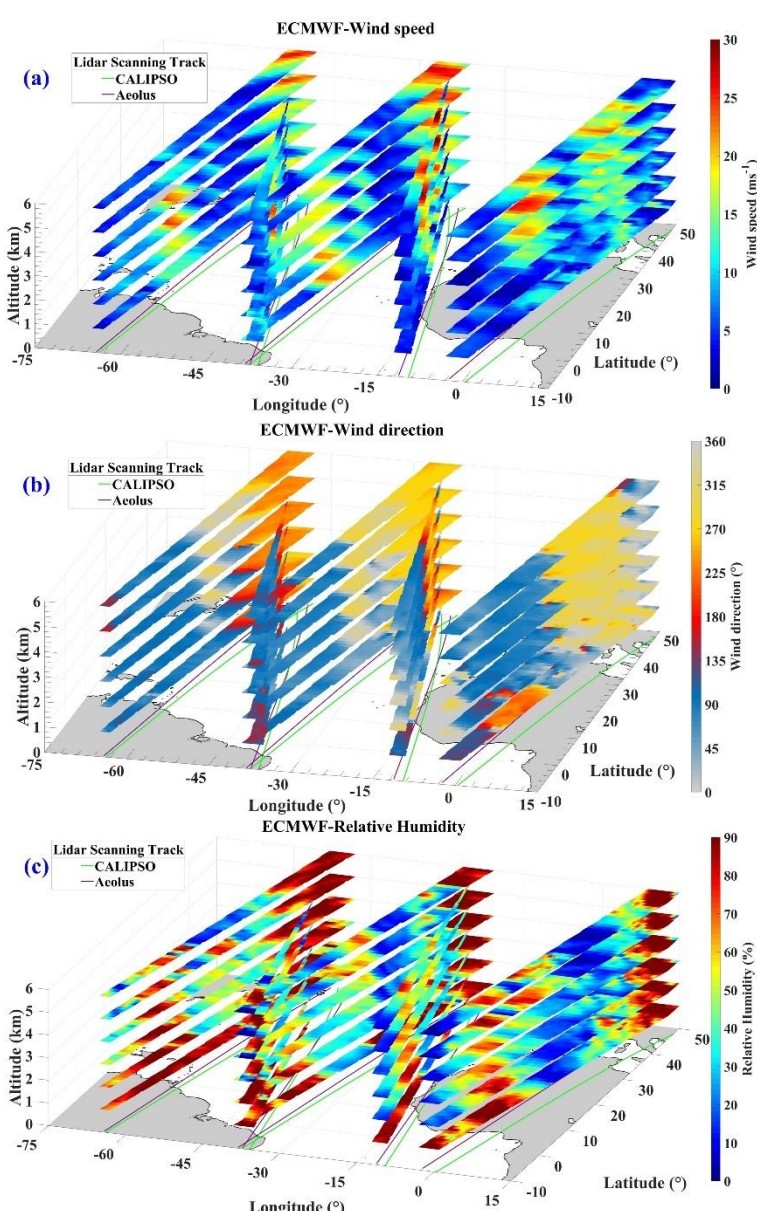

**Figure 9. The wind speed, wind directions and relative humidty between the quasi-synchronization observation tracks of Aeolus and CALIPSO provided by ECMWF.**



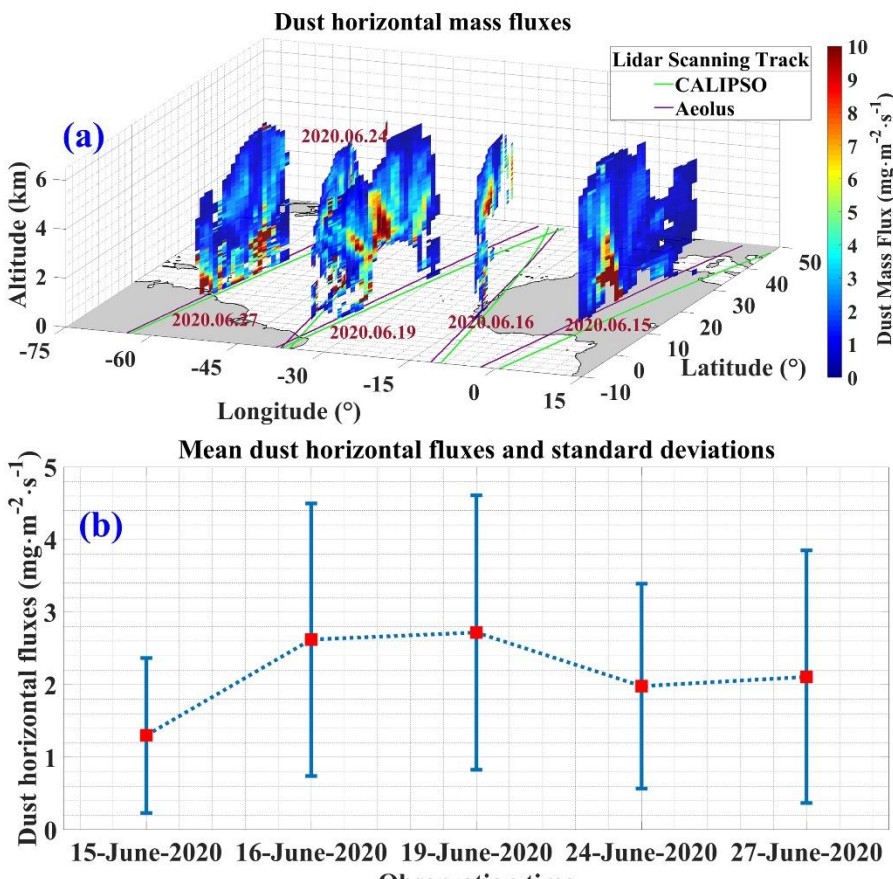

**Figure 10. The dust horizontal fluxes calculated with data from ALADIN, CALIOP and ECMWF (a) at different cross-sections and at different times during the dust transport and (b) the corresponding mean dust fluxes and standard deviations.**

In Figure 10, the dust horizontal fluxes at different heights of all the cross-sections during the dust transport, as well as the dust flux distributions are presented. The trend of the dust fluxes is also shown in Figure 10(b) and the dust horizontal fluxes are about $1.30 \pm 1.07$ mg$\cdot$m$^{-2}\cdot$s$^{-1}$ on 15 June 2020, $2.62 \pm 1.88$ mg$\cdot$m$^{-2}\cdot$s$^{-1}$ on 16 June 2020, $2.72 \pm 1.89$ mg$\cdot$m$^{-2}\cdot$s$^{-1}$ on 19 June 2020, $1.98 \pm 1.41$ mg$\cdot$m$^{-2}\cdot$s$^{-1}$ on 24 June 2020 and $2.11 \pm 1.74$ mg$\cdot$m$^{-2}\cdot$s$^{-1}$ on 27 June 2020. From this trend, it is illuminated that the flux value is the smallest (around $1.30 \pm 1.07$ mg$\cdot$m$^{-2}\cdot$s$^{-1}$) when the dust is initially generated on 15 June. With the development and enhancement of the dust event, the flux value gradually increases and reaches the peak value (around $2.72 \pm 1.89$ mg$\cdot$m$^{-2}\cdot$s$^{-1}$) on 19 June. Then, the flux value gradually decreases until the dust deposited in the west part of Atlantic Ocean, the Americas and the Caribbean Sea.



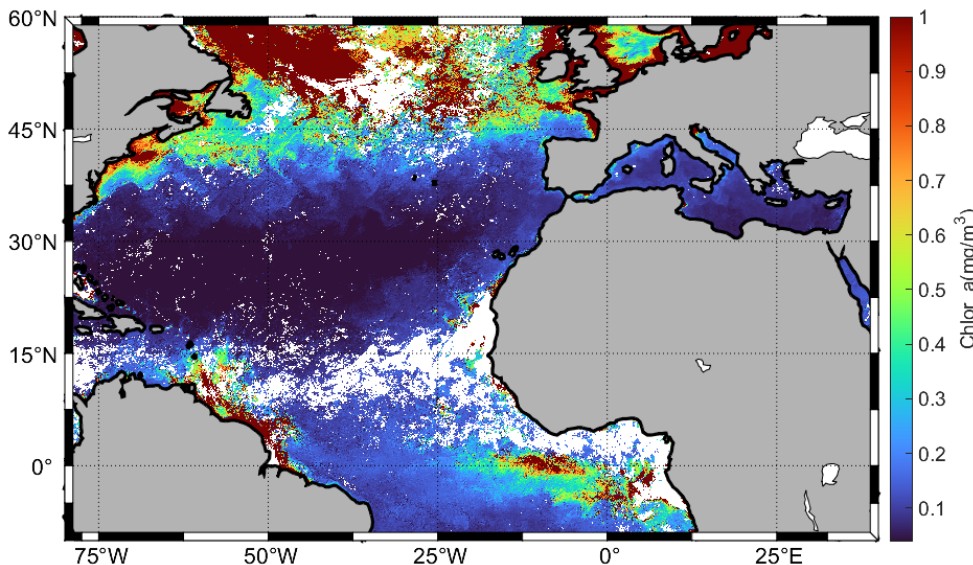


**Figure 11. The monthly Chlorophyll concentration in June 2020 measured with MODIS-Aqua.**

In Figure 11, the monthly Chlorophyll concentrations at the zone of 10°S~60°N and 75°W~30°E from in June 2020 measured with MODIS-Aqua are provided. The data is obtained via the NASA ocean color website (https://oceancolor.gsfc.nasa.gov/data/overview/, last access on the 5$^{th}$ March 2021). From this figure, it can be seen that the

Chlorophyll concentrations at the zone of 10°N~45°N and 75°W~30°E is rather low (except the northeast coast of South America) and hence the whole transport route of the dust event are mainly over the oligotrophic regions at the Atlantic Ocean, the Americas and the Caribbean Sea. The mineral dust, probably act as the main nutrient source, delivers micronutrients including soluble Fe and P to the deposition zones and has the potential to fertilize the ocean and increase the primary productivity in the Atlantic Ocean and Caribbean Sea. And thus leading to $N_2$ fixation and $CO_2$ drawdown.

From Figure 12, the L2C wind vectors including u and v components from Aeolus at different times are plotted. In Figure 12(a), the dust plumes are trapped in the Northeasterly Trade-wind zone (indicated by the blue colour at different cross-sections) between the latitudes of 5 °N and 30 °N and altitudes of 0 km and 6 km. The u component values of the wind vectors in the Trade-wind zone are high and reach to 20 m·s$^{-1}$. Dominated by the Trade-wind, the dust plumes are mainly transported to the west direction. This space area looks like a tunnel and the dust plumes are transported inside. Since the dust plumes are

frequently occurred in this area, this tunnel can be called as "Saharan dust eastward transport tunnel". From Figure 12(b), the v component values of the wind vectors are presented as well. Effected by the small wind towards south direction, the dust plumes are slight shifted to the south part of Atlantic Ocean in this case.





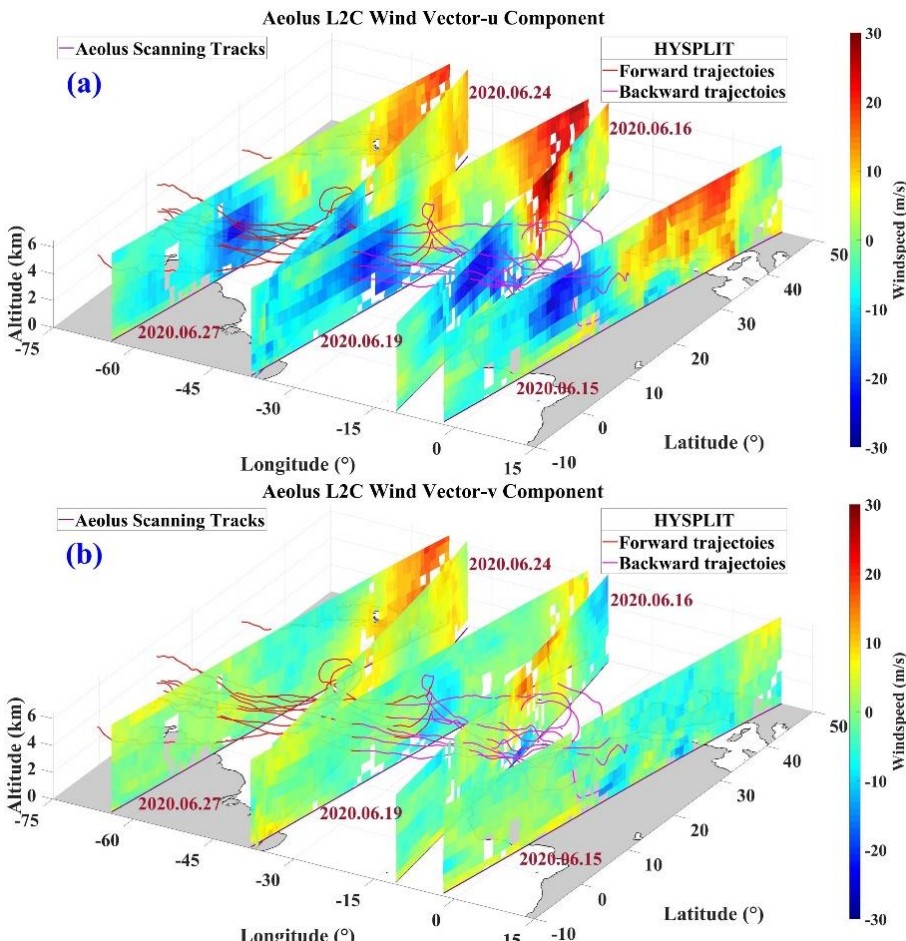

**Figure 11. The u and v components provided by Aeolus and the HYSPLIT model for the dust event.**

**5. Summary and conclusions**

In this study, a long-term large-scale Sahara dust transport event occurred during 14 June and 27 June 2020 is tracked and its horizontal fluxes are calculated with the remote measurement data from ALADIN, CALIOP and the reanalysis data from ECMWF and HYSPLIT. The combination of the spaceborne lidars ALADIN and CALIOP measurement data coupled with the products from ECMWF and HYSPLIT schemes to (1) evaluate the performance of the ALADIN and CALIOP on the observations of dust optical properties and wind fields and (2) explore the capability in tracking the dust events and in calculating the dust horizontal mass fluxes.

We identified the dust plumes with AIRS/Aqua Dust Score Index and with the Vertical Feature Mask products from CALIOP. The emission, dispersion, transport and deposition of the dust event are followed using the data from HYSPLIT, CALIOP and AIRS/Aqua. With the quasi-synchronization observations by ALADIN and CALIOP, combining the wind



vectors and relative humidity, the dust horizontal fluxes are calculated. The accuracy improvement of dust horizontal flux estimations with the complement of Aeolus-produced aerosol optical properties and wind data is foreseen when compared with the traditional assessments based on the data from CALIOP and models.

From this study, it is found that the dust event generated on 14 and 15 June 2020 from Sahara Desert in North Africa, and then dispersed and transported westward over the Atlantic Ocean, and finally deposited in the west part of Atlantic Ocean, the

Americas and the Caribbean Sea. During the transport and deposition processes, the dust plumes are trapped and transport in the Northeasterly Trade-wind zone between the latitudes of 5 °N and 30 °N and altitudes of 0 km and 6 km (we name this space area as "Saharan dust eastward transport tunnel"). From the measurement results on 19 June 2020, influenced by the hygroscopic effect and mixing with other types aerosols, the backscatter coefficients of dust plumes are increasing along the transport routes, with $3.88 \times 10^{-6} \pm 2.59 \times 10^{-6}$ $m^{-1}sr^{-1}$ in "dust portion during emission phase", $7.09 \times 10^{-6} \pm 3.34 \times 10^{-6}$ $m^{-1}sr^{-1}$

in "dust portion during development phase" and $7.76 \times 10^{-6} \pm 3.74 \times 10^{-6}$ $m^{-1}sr^{-1}$ in "dust portion during deposition phase".

Finally, the horizontal fluxes at different dust parts and heights on 19 June and on entire transport routine during transportation are computed, as shown in Figure 7 and Figure 10, respectively. On 19 June, the dust horizontal fluxes are about $2.17 \pm 1.83$ $mg \cdot m^{-2} \cdot s^{-1}$ in dust portion during emission phase, $2.72 \pm 1.89$ $mg \cdot m^{-2} \cdot s^{-1}$ in dust portion during development phase and $3.01 \pm 2.77$ $mg \cdot m^{-2} \cdot s^{-1}$ in dust portion during deposition phase. In the whole life-time of the dust event, the dust

horizontal fluxes are about $1.30 \pm 1.07$ $mg \cdot m^{-2} \cdot s^{-1}$ on 15 June 2020, $2.62 \pm 1.88$ $mg \cdot m^{-2} \cdot s^{-1}$ on 16 June 2020, $2.72 \pm 1.89$ $mg \cdot m^{-2} \cdot s^{-1}$ on 19 June 2020, $1.98 \pm 1.41$ $mg \cdot m^{-2} \cdot s^{-1}$ on 24 June 2020 and $2.11 \pm 1.74$ $mg \cdot m^{-2} \cdot s^{-1}$ on 27 June 2020. It is learned from this study, the minimum of the fluxes appears when the dust event is initially generated on 15 June. During the dust development stage, the horizontal fluxes gradually increase and reach to the maximum value on 19 June with the enhancement of the dust event. Then, the horizontal fluxes gradually decrease since most of the dust deposited in the

Atlantic Ocean, the Americas and the Caribbean Sea. Combining the Chlorophyll concentrations records from MODIS-Aqua, the Saharan Dust is found transported across the oligotrophic regions Atlantic Ocean towards the Americas and Caribbean Sea, which are also oligotrophic regions. The mineral dust delivers micronutrients including soluble Fe and P to the deposition zones and has the potential to fertilize the ocean and increases the primary productivity in the Atlantic Ocean and Caribbean Sea. And thus leading to $N_2$ fixation and $CO_2$ drawdown. As a further research, the impact of dust transport and fertilization

on the biological communities in eutrophic and oligotrophic areas is of great significance. Additionally, under the combined influence of the Western Boundary Current (e.g., Gulf stream), the impact of dust deposition on the Caribbean Sea and the North Atlantic Ocean (such as the occurrence of brown algae blooms) is also worth studying.

**Data availability.**

The Aeolus data are obtained from the tool VirES for Aeolus (https://aeolus.services/, last accessed on 19 February 2021). The

CALIOP data can be downloaded from https://eosweb.larc.nasa.gov/project/CALIPSO (last accessed on 19 February 2021).



The ECMWF reanalysis ERA5 wind data can be accessed from https://cds.climate.copernicus.eu/cdsapp#!/dataset/reanalysis-era5-pressure-levels?tab=form (last accessed on 19 February 2021). The backward trajectory and forward trajectory of HYSPLIT can be run at https://www.ready.noaa.gov/HYSPLIT_traj.php (last accessed on 19 February 2021). Chlorophyll concentrations at the zone of   and   from 17 June 2020 to 24 June 2020 measured with MODIS-Aqua are provided. The data

is obtained via the NASA ocean color website (https://oceancolor.gsfc.nasa.gov/data/overview/, last access on 5 March 2021).

**Author contributions.**

G. Dai and S. Wu conceived of the idea for the dust transport and horizontal fluxes measurement with spaceborne lidars ALADIN, CALIOP and model reanalysis data; G. Dai wrote the manuscript; K. Sun, G. Dai, S. Wu and B. Liu conducted the data analyses; X. Wang helped in programming, X. E and Q. Liu downloaded the ECMWF and MODIS data, and all the co-

authors discussed the results and reviewed the manuscript.

**Competing interests.**

The authors declare that they have no conflict of interest.

**Special issue statement.**

This article is part of the special issue "Aeolus data and their application". It is not associated with a conference.

**Acknowledgments.**

This study has been jointly supported by the National Key Research and Development Program of China under grant 2019YFC1408001 and 2019YFC1408002 and the National Natural Science Foundation of China (NSFC) under grant 41905022 and 61975191.

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
