# Peer review of "Dust transport and horizontal fluxes measurement with spaceborne lidars ALADIN, CALIOP and model reanalysis data"

_Atmospheric Chemistry and Physics, 2021_

## Author Comment (AC1)

Response to Reviewer #1:

Overall comments:

The paper presents a large event of dust transport originating from the Sahara and crossing the tropical Atlantic ocean westward, until it reaches Central America and the Carribean sea.

This event was large in the amount of dust transported and was reported in newspapers (e.g. https://www.nytimes.com/2020/06/22/science/saharan-dust-plume.html). Such a large plume is an easy target for Aeolus detection of aerosols and is one of the first uses of this High Spectral Resolution Lidar optical properties product. For both these aspects, this paper makes an interesting contribution to the observation of the atmosphere and is relevant to the journal scope.

The structure of the paper is clear. The syntax and grammar are understandable but will require an extensive work to make it easier to read.

AR: Thanks. We have polished the English language throughout the whole revised manuscript to make it easier to read.

The event is large enough that the transported dust is visible without a sophisticated analysis. However, the method used to match CALIPSO and Aeolus data, to estimate the mass flux and the associated errors needs to be described in more details. Many mathematical details are missing to provide a complete understanding of the work. The flow charts give a good overview of the procedure but are not enough.

AR: Thank you for your suggestion.

For the method used to match CALIPSO and Aeolus data:

Since the orbits of Aeolus and CALIPSO are different, they cannot meet each other at the exactly same time and same location. In our study, the closest CALIPSO scanning tracks to those of Aeolus, are about 4 hours ahead of Aeolus. Based on the transport directions of dust events modelled with HYSPLIT, the tracks of Aeolus should be

always downwind of the tracks of CALIPSO. When the tracks of Aeolus and CALIPSO are selected, the distances between the tracks can be calculated. Assuming that the wind speed between CALIPSO scanning tracks and Aeolus scanning tracks is in the range of 5m·s⁻¹ to 15m·s⁻¹, the corresponding transport distances of the dust plumes are in the range of 72km to 216km. Besides, during the short-time transportation of Sahara dust plume, dust optical properties (volume linear depolarization ratio at 532 nm) maintain almost unchanged (Haarig et al., 2017).

[Figure]

Figure 1. Saharan air layer (from 1.2 to 4.2 km height) above the marine boundary layer. Panel (b) shows the range-corrected cross-polarized backscatter signal at 532 nm with temporal resolution of 30 s and vertical resolution of 7.5 m, respectively. Panel (d) shows the volume linear depolarization ratio at 532 nm. The radiosonde profiles of wind speed (WS) and wind direction (WD) are shown in panel (a), the profiles of relative humidity (RH) and temperature (T) in panel (c). The lidar observation was performed on 6 July 2014, 23:18–01:33 UTC. The radiosonde was launched at

23:47 UTC (indicated by a black vertical line). [Figure 11 from Haarig et al. (2017).]

Figure 1 shows the wind direction, wind speed, relative humidity and temperature measured by radiosonde, as well as the range-corrected cross-polarized Sahara dust backscatter signal at 532 nm and the Sahara dust volume linear depolarization ratio at 532 nm measured by BERTHA (Backscatter Extinction lidar-Ratio Temperature Humidity profiling Apparatus) at Barbados, which is located in Eastern Caribbean (Haarig et al., 2017). It can be seen from Figure 1 that even after long-range transportation, the optical properties (volume linear depolarization ratio at 532 nm) of the dust layer maintain stable in about 2 hours. Therefore, in our work, we think that during the short-time (~4 hours) transportation of Sahara dust plume, dust optical properties maintain almost unchanged.

Consequently, in our study, if the distances between two satellites scanning tracks are less than 200 km and the tracks of Aeolus are downwind of the tracks of CALIPSO, it is reasonable to state that the dust plumes captured by CALIPSO are transported towards the Aeolus scanning regions in around 4 hours, hence the following procedures could be continued. From the measurement cases provided by the paper mentioned above, the optical properties of Sahara dust within short-time (e.g., 2h, 4h…) seems to be stable. Hence, we insist that the combination of the optical properties provided by CALIPSO and Aeolus, with 4 hours difference, is applicable/reasonable to calculate the volume concentration and mass concentration.

To make it clear to the reader, we also add the similar explanations in the Section 3 of the revised manuscript as following:

"Assuming the wind speed between CALIPSO scanning tracks and Aeolus scanning tracks is in the range of 5 $m \cdot s^{-1}$ to 15 $m \cdot s^{-1}$, the transport distances of the dust plumes are in the range of 72km to 216km. Besides, during the short-time transportation of Sahara dust plume, dust optical properties maintain almost unchanged (Haarig et al., 2017). Consequently, in our study, if the distances between two satellites scanning tracks are less than 200 km and the tracks of Aeolus are downwind of the tracks of CALIPSO, it is reasonable to state that the dust plumes captured by CALIPSO are transported towards the Aeolus scanning regions in around 4 hours, hence the following

procedures could be continued."

For estimating the mass flux and the associated errors:

Firstly, it should be emphasized that, in the revised manuscript, we define "dust advection" instead of "mass flux" to describe dust transportation quantificationally. The "dust advection" is the multiplication of the mass concentration (m) and the horizontal wind velocity (v), which means it is a vector.

For estimating mass concentration of dust aerosol, firstly, the regularization method reported in Müller et al. (1999) is applied to calculate the dust aerosol volume concentration. In our study, three extinction coefficients (at 355nm, 532nm and 1064nm respectively) and two backscatter coefficients (at 532nm and 1064nm respectively) are needed to compute the volume concentration. The advantage of this method is that it does not require prior knowledge of the shape of the particle size distribution and the measurement estimate uncertainty of aerosol volume concentration is on the order of 50% if the estimated errors of the inputs are on the order of 20%. The input parameters of regularization method in this work are the backscatter coefficient and extinction coefficient at 532nm and 1064 nm from CALIPSO, the extinction coefficient at 355 nm from Aeolus.

For the accuracy of the CALIPSO-retrieved extinction and backscatter coefficient, as stated/reported in Getzewich et al. (2018): "Extensive validation data acquired by NASA's airborne high spectral resolution lidar (HSRL) shows that during the daytime the average difference between collocated CALIPSO and HSRL measurements of 532 nm attenuated backscatter coefficients is reduced from 3.3%±3.1 % in V3 to 1.0%±3.5 % in V4.". In Vaughan et al. (2019): "By evaluating calibration coefficients derived using both water clouds and ocean surfaces as alternate calibration targets, and through comparisons to independent, collocated measurements made by airborne high spectral resolution lidar, we conclude that the CALIOP V4 1064 nm calibration coefficients are accurate to within 3 %.". And in Kim et al. (2018): "The uncertainty in the V4 dust lidar ratio of 20 % (30 %) at 532 nm (1064 nm) accounts for the regional variability.". It is considered that the estimate errors of backscatter coefficients and extinction coefficients from CALIPSO are on the order of 20%.

For the accuracy of the Aeolus-retrieved extinction coefficient, thank to your publication entitled "Aeolus L2A Aerosol Optical Properties Product: Standard Correct Algorithm and Mie Correct Algorithm" in ATMD, it is very significant for us to acquire more information about Aeolus L2A product (Flament et al, 2021). Figure 2 shows the inputs and simulation results simulated by the Aeolus End-To-End Simulator. It can be seen that the simulation extinction coefficients fit the inputs well mostly, especially when the altitude is larger than 2km. Besides, in terms of quality control, negative extinction coefficient values of L2A are excluded while the "bin_1_clear" flag and the "processing_qc_flag" of L2A are used to eliminate invalid data.

[Figure]

Figure 2. Backscatter (Top) and Extinction profiles (Bottom) comparison between E2S simulations processed up to the SCA mid-bin product and the E2S inputs. The black line is the simulation input parameters averaged within each BRC. Thin black line represents the associated standard deviation. Red profiles are the mean coefficients retrieved by the SCA algorithm from 20 realizations. Lighter shading is the associated standard deviation i.e. the true variability. Darker shading delimits the mean error estimated by the SCA error propagation algorithm. [Figure 10 from Flament et al. (2021)]

In conclusion, we think that the estimated errors of the five input parameters we

used to calculate the aerosol volume concentration are on the order of 20%. Hence, the error of the volume concentration computed with the regularization method should be on the order of 50%.

After estimating aerosol volume concentration, an assuming typical dust particle density which is set as 2.65 g·cm$^{-3}$ referring to previous studies (e.g., Schepanski et al., 2009; Hofer et al., 2017; Mamouri and Ansmann, 2017), the particle mass concentration would be estimated as Engelmann et al. (2008) introduced. Finally, by multiplying mass concentration (m) and horizontal wind velocity from Aeolus L2C, dust advection is calculated. The estimate errors of dust advection are the combination of mass concentration estimate errors (~50%) and Aeolus L2C wind vector estimate errors.

*Reference:*

*Engelmann, R., Wandinger, U., Ansmann, A., Müller, D., Žeromskis, E., Althausen, D., and Wehner, B.: Lidar observations of the vertical aerosol flux in the planetary boundary layer, J. Atmos. Ocean. Tech., 25, 1296-1306, https://doi.org/10.1175/2007JTECHA967.1, 2008.*

*Flament, T., Trapon, D., Lacour, A., Dabas, A., Ehlers, F., and Huber, D.: Aeolus L2A Aerosol Optical Properties Product: Standard Correct Algorithm and Mie Correct Algorithm, Atmos. Meas. Tech. Discuss. [preprint], https://doi.org/10.5194/amt-2021-181, in review, 2021.*

*Getzewich, B. J., Vaughan, M. A., Hunt, W. H., Avery, M. A., Powell, K. A., Tackett, J. L., Winker, D. M., Kar, J., Lee, K.-P., and Toth, T. D.: CALIPSO lidar calibration at 532 nm: version 4 daytime algorithm, Atmos. Meas. Tech., 11, 6309–6326, https://doi.org/10.5194/amt-11-6309-2018, 2018.*

*Haarig, M., Ansmann, A., Althausen, D., Klepel, A., Groß, S., Freudenthaler, V., Toledano, C., Mamouri, R.-E., Farrell, D. A., Prescod, D. A., Marinou, E., Burton, S. P., Gasteiger, J., Engelmann, R., and Baars, H.: Triple-wavelength depolarization-ratio profiling of Saharan dust over Barbados during SALTRACE in 2013 and 2014, Atmos. Chem. Phys., 17, 10767–10794, https://doi.org/10.5194/acp-17-10767-2017, 2017.*

*Hofer, J., Althausen, D., Abdullaev, S. F., Makhmudov, A. N., Nazarov, B. I., Schettler, G., Engelmann, R., Baars, H., Fomba, K. W., and Müller, K.: Long-term profiling of mineral dust and pollution aerosol with multiwavelength polarization Raman lidar at the Central Asian site of Dushanbe, Tajikistan: case studies, Atmos. Chem. Phys., 17, 14559-14577, https://doi.org/10.5194/acp-17-14559-2017, 2017.*

*Kim, M.-H., Omar, A. H., Tackett, J. L., Vaughan, M. A., Winker, D. M., Trepte, C. R., Hu, Y., Liu, Z., Poole, L. R., Pitts, M. C., Kar, J., and Magill, B. E.: The CALIPSO version 4 automated aerosol classification and lidar ratio selection algorithm, Atmos. Meas. Tech., 11, 6107–6135, https://doi.org/10.5194/amt-11-6107-2018, 2018.*

*Mamouri, R.-E., and Ansmann, A.: Potential of polarization/Raman lidar to separate fine dust, coarse dust, maritime, and anthropogenic aerosol profiles, Atmos. Meas. Tech., 10, 3403-3427, https://doi.org/10.5194/amt-10-3403-2017, 2017.*

*Müller, D., Wandinger, U., and Ansmann, A.: Microphysical particle parameters from extinction and backscatter lidar data by inversion with regularization: theory, Appl. Opt., 38, 2346-2357, https://doi.org/10.1364/AO.38.002346, 1999.*

*Schepanski, K., Tegen, I., and Macke, A.: Saharan dust transport and deposition towards the tropical northern Atlantic, Atmos. Chem. Phys., 9, 1173-1189, https://doi.org/10.5194/acp-9-1173-2009, 2009.*

*Vaughan, M., Garnier, A., Josset, D., Avery, M., Lee, K.-P., Liu, Z., Hunt, W., Pelon, J., Hu, Y., Burton, S., Hair, J., Tackett, J. L., Getzewich, B., Kar, J., and Rodier, S.: CALIPSO lidar calibration at 1064 nm: version 4 algorithm, Atmos. Meas. Tech., 12, 51–82, https://doi.org/10.5194/amt-12-51-2019, 2019.*

We need a reference about the L2A processor. I would suggest Flamant et al. (2008, https://doi.org/10.1111/j.1600-0870.2007.00287.x). Alternatively, we, the L2A development team, are currently preparing about the current L2A product which will be submitted soon to the Atmospheric Measurement Techniques journal. It will present up to date information about the product.

AR: Thank you for your suggestion. Flamant et al. (2008,

https://doi.org/10.1111/j.1600-0870.2007.00287.x) is added in the section 2.1 of revised manuscript to demonstrate L2A processor. Besides, we are very glad to see the publication of your work "Aeolus L2A Aerosol Optical Properties Product: Standard Correct Algorithm and Mie Correct Algorithm" in AMTD (Flament et al, 2021). It's quite significant for us to refer to your work for more information about Aeolus L2A product.

The relevant references are added in the revised manuscript and are listed below, to describe the L2A processor and the L2A product:

Flamant, P., Cuesta, J., Denneulin, M. L., Dabas, A., and Huber, D.: ADM-Aeolus retrieval algorithms for aerosol and cloud products. Tellus A: Dynamic Meteorology and Oceanography, 60(2), 273-286. https://doi.org/10.1111/j.1600-0870.2007.00287.x, 2008.

Flament, T., Trapon, D., Lacour, A., Dabas, A., Ehlers, F., and Huber, D.: Aeolus L2A Aerosol Optical Properties Product: Standard Correct Algorithm and Mie Correct Algorithm, Atmos. Meas. Tech. Discuss. [preprint], https://doi.org/10.5194/amt-2021-181, in review, 2021.

Also, the Aeolus L2A dataset was not widely used yet: more information needs to be provided about the content and overall data quality of the product. Additionally, a discussion of the quality controls applied by the authors would be welcome.

AR: Thank you for your suggestion. In terms of Aeolus L2A quality control, negative backscatter and extinction coefficient values of L2A are excluded. And the "bin_1_clear" flag and the "processing_qc_flag" of L2A are used to eliminate invalid data. The detailed information can be found vis the website of https://stcorp.github.io/codadef-documentation/AEOLUS/types/Level_2A_SCA_PCD_ADSR_03_12.html (last accessed on 21 July 2021).

Additionally, the quality control information for the input parameters are also added and highlighted in Figure 2 in the revised manuscript. You can also refer to this

figure below:

[Figure]

On a technical aspect of data analysis: we recommend to use the "mid bin" product of the SCA. The authors do not mention the existence of two products in the SCA. This is especially important when looking at extinction coefficients, which are more sensitive to noise and better retrieved through this averaged version of the algorithm.

As few papers are available on the subject, this is a technical detail that needs to be mentioned. See also Baars et al. (2021, https://doi.org/10.1029/2020GL092194, "As for the extinction, the SCA mid bin algorithm is to be preferred against the SCA for the lidar ratio analysis.").

AR: Thanks. According to your suggestion, we cite Baars et al. (2021) and Flament et al.(2021), supplementing the description of the "mid bin" product of the SCA in detail in section 3 of revised manuscripts, as following:

"Additionally, we use the mid bin product (sca_optical_properties_mid_bins) of SCA instead of the normal product of SCA, because the mid-bin algorithm provides more robust results (Baars et al., 2021; Flament et al., 2021). The extinction coefficients, which are more sensitive to noise and are the significant inputs of the dust advection calculation, are better retrieved through this "mid bin" averaged version of the algorithm. In terms of quality control, negative extinction coefficient values of L2A are excluded while the "bin_1_clear" flag and the "processing_qc_flag" of L2A are used to eliminate invalid data."

In the revised manuscript, the "mid bin" extinction coefficient product of the SCA are used instead of the original dataset in the former manuscript. As shown in the figures below, the extinction coefficient can provide significant information. It makes our calculation much more solid.

[Figure]

Reference:

Baars, H., Radenz, M., Floutsi, A. A., Engelmann, R., Althausen, D., Heese, B., ... & Wandinger, U.: Californian wildfire smoke over Europe: A first example of the aerosol observing capabilities of Aeolus compared to ground-based lidar. Geophysical

*Research Letters, 48(8), e2020GL092194. https://doi.org/10.1029/2020GL092194, 2021.*

*Flament, T., Trapon, D., Lacour, A., Dabas, A., Ehlers, F., and Huber, D.: Aeolus L2A Aerosol Optical Properties Product: Standard Correct Algorithm and Mie Correct Algorithm, Atmos. Meas. Tech. Discuss. [preprint], https://doi.org/10.5194/amt-2021-181, in review, 2021.*

Detailed line by line comments:

l. 145-147: It is right to use the SCA rather than the ICA (this is what we recommend as developers of the product).

However, as said above, the SCA provides two sets of extinction and backscatter coefficients. If the extinction is needed, I recommend using the "mid bin" solution. If this was not the case in the current analysis, this might change results significantly. Known defects of the SCA algorithm will be discussed in the future paper we are preparing. In the mean time, I am available to discuss this directly if necessary.

AR: In our study, as a significant input parameter, the extinction coefficient at 355 nm from Aeolus L2A products is used to estimate the dust mass concentration. Therefore, the more robust and accurate "mid bin" solution of L2A products make the analysis and calculation of this work more credible. In the revised manuscript, Aeolus L2A "mid bin" products are used to analyze the dust transportation and used to estimate the dust mass concentration. The corresponding statement in the Section 3 of revised manuscript is organized as below:

"Additionally, we use the mid bin product (sca_optical_properties_mid_bins) of SCA instead of the normal product of SCA, because the mid-bin algorithm provides more robust results (Baars et al., 2021; Flament et al., 2021). The extinction coefficients, which are more sensitive to noise and are the significant inputs of the dust advection calculation, are better retrieved through this "mid bin" averaged version of the algorithm. In terms of quality control, negative extinction coefficient values of L2A

are excluded while the "bin_1_clear" flag and the "processing_qc_flag" of L2A are used to eliminate invalid data."

l. 152: "Since the footprints of Aeolus and CALIPSO are not exactly matched, the missing wind data between their tracks have to be filled in using the ERA5 wind field data". This sentence is very unclear. Does that mean that you use ERA5 winds at the location of the CALIPSO track? This whole "Methodology" section needs to be rewritten with much more detail.

AR: Thanks. There are two purposes on the usage the ERA5 wind field data between Aeolus and CALIPSO tracks. One is that the ERA5 wind speed and direction data provide the evidence of dust transporting from CALIPSO tracks towards Aeolus tracks. Besides, as shown in the figures below, the ERA5 wind field data between the tracks of Aeolus and CALIPSO at all height surfaces are smoothly distributed and the values are stable. It means that the Aeolus L2C data can be used at the location of the CALIPSO track. The corresponding description and explanation are supplemented in the section 3 of the revised manuscripts.

[Figure]

[Figure]

We rewrite the "Methodology" section in the revised manuscripts, which include detailed description of track-match method, datasets and quality control method, data processing and dust advection calculation.

l. 153: "and the measurement uncertainty is on the order of 20%." Which measurement uncertainty is discussed here? Aeolus estimated errors are often larger than 20 %.

AR: Thanks. Sorry for the misleading. According to Müller et al. (1999), by using regularization method which was performed by generalized cross-validation (GCV), the estimate uncertainty of aerosol volume concentration is on the order of 50% if the estimated errors of the input parameters are on the order of 20%. The inputs in this work are backscatter coefficient and extinction coefficient at 532 nm, backscatter coefficient and extinction coefficient at 1064 nm from CALIPSO and extinction coefficient at 355 nm from Aeolus.

As described above, for the accuracy of the CALIPSO-retrieved extinction and backscatter coefficients: For the backscatter coefficient at 532 nm, during the daytime, the average difference between collocated CALIPSO and HSRL measurements is 1.0%±3.5 % in V4 (Getzewich et al., 2018); for the backscatter coefficient at 1064 nm, by evaluating calibration coefficients derived using both water clouds and ocean surfaces as alternate calibration targets, and through comparisons to independent,

collocated measurements made by airborne high spectral resolution lidar, the CALIOP V4 1064 nm calibration coefficients are accurate to within 3 % (Vaughan et al., 2019); for the extinction coefficients, the uncertainty in the V4 dust lidar ratio of 20 % (30 %) at 532 nm (1064 nm) (Kim et al., 2018), thus it is considered that the estimate errors of the extinction coefficients from CALIPSO are on the order of 20%. Consequently, we insist that the uncertainties of CALIPSO-retrieved extinction and backscatter coefficients are on the order of 20%.

For the accuracy of the Aeolus-retrieved extinction coefficient, it is considered that the simulation extinction coefficients fit the inputs well mostly, especially when the altitude is larger than 2km (Flament et al, 2021). Besides, in terms of quality control, negative extinction coefficient values of L2A are excluded while the "bin_1_clear" flag and the "processing_qc_flag" of L2A are used to eliminate invalid data. Hence, we insist that Aeolus L2A extinction coefficient can be the input parameters of the regularization method.

In conclusion, we think that the estimated errors of the five input parameters we used to calculate the aerosol volume concentration are on the order of 20%. Hence, the error of the volume concentration computed with the regularization method should be on the order of 50%.

The corresponding correction and description are addressed in Section 3.3 of the revised manuscripts.

*Reference:*

*Flament, T., Trapon, D., Lacour, A., Dabas, A., Ehlers, F., and Huber, D.: Aeolus L2A Aerosol Optical Properties Product: Standard Correct Algorithm and Mie Correct Algorithm, Atmos. Meas. Tech. Discuss. [preprint], https://doi.org/10.5194/amt-2021-181, in review, 2021.*

*Getzewich, B. J., Vaughan, M. A., Hunt, W. H., Avery, M. A., Powell, K. A., Tackett, J. L., Winker, D. M., Kar, J., Lee, K.-P., and Toth, T. D.: CALIPSO lidar calibration at 532 nm: version 4 daytime algorithm, Atmos. Meas. Tech., 11, 6309–6326, https://doi.org/10.5194/amt-11-6309-2018, 2018.*

*Kim, M.-H., Omar, A. H., Tackett, J. L., Vaughan, M. A., Winker, D. M., Trepte, C. R., Hu, Y., Liu, Z., Poole, L. R., Pitts, M. C., Kar, J., and Magill, B. E.: The CALIPSO version 4 automated aerosol classification and lidar ratio selection algorithm, Atmos. Meas. Tech., 11, 6107–6135, https://doi.org/10.5194/amt-11-6107-2018, 2018.*

*Müller, D., Wandinger, U., and Ansmann, A.: Microphysical particle parameters from extinction and backscatter lidar data by inversion with regularization: theory, Appl. Opt., 38, 2346-2357, https://doi.org/10.1364/AO.38.002346, 1999.*

*Vaughan, M., Garnier, A., Josset, D., Avery, M., Lee, K.-P., Liu, Z., Hunt, W., Pelon, J., Hu, Y., Burton, S., Hair, J., Tackett, J. L., Getzewich, B., Kar, J., and Rodier, S.: CALIPSO lidar calibration at 1064 nm: version 4 algorithm, Atmos. Meas. Tech., 12, 51–82, https://doi.org/10.5194/amt-12-51-2019, 2019.*

l. 160:

- References are provided regarding the determination of aerosol mass (namely, Müller et al. 1999) but a discussion about the method is needed. For instance, Müller et al. say the algorithm can cope with errors of ~20% on the lidar data. Is this verified here?

AR: Thank you for your suggestion. The method of aerosol volume concentration calculation is introduced and discussed in detail in Müller et al. (1999). Following the method introduced in Müller et al. (1999), we used backscatter coefficient and extinction coefficient at 532 nm, backscatter coefficient and extinction coefficient at 1064 nm from CALIPSO, and extinction coefficient at 355 nm from Aeolus to estimate dust volume concentration. And according to previous studies (e.g., Schepanski et al., 2009; Hofer et al., 2017; Mamouri and Ansmann, 2017), the assuming typical dust density is set as 2.65 g·cm$^{-3}$ to multiply with dust volume concentration and acquire mass concentration.

The estimate uncertainty of aerosol volume concentration by regularization method is on the order of 50%, not 20%.

As mentioned above, for the accuracy of the CALIPSO-retrieved extinction and

backscatter coefficients: For the backscatter coefficient at 532 nm, during the daytime, the average difference between collocated CALIPSO and HSRL measurements is 1.0%±3.5 % in V4 (Getzewich et al., 2018); for the backscatter coefficient at 1064 nm, the CALIOP V4 1064 nm calibration coefficients are accurate to within 3 % (Vaughan et al., 2019); for the extinction coefficients, the uncertainty in the V4 dust lidar ratio of 20 % (30 %) at 532 nm (1064 nm) (Kim et al., 2018), thus it is considered that the estimate errors of the extinction coefficients from CALIPSO are on the order of 20%. Consequently, we insist that the uncertainties of CALIPSO-retrieved extinction and backscatter coefficients are on the order of 20%.

For the accuracy of the Aeolus-retrieved extinction coefficient, it is considered that the simulation extinction coefficients fit the inputs well mostly, especially when the altitude is larger than 2km (Flament et al, 2021). Besides, in terms of quality control, negative extinction coefficient values of L2A are excluded while the "bin_1_clear" flag and the "processing_qc_flag" of L2A are used to eliminate invalid data. Hence, we insist that Aeolus L2A extinction coefficient can be the input parameters of the regularization method.

In conclusion, we think that the estimated errors of the five input parameters we used to calculate the aerosol volume concentration are on the order of 20%. Hence, the error of the volume concentration computed with the regularization method should be on the order of 50%.

Reference:

Getzewich, B. J., Vaughan, M. A., Hunt, W. H., Avery, M. A., Powell, K. A., Tackett, J. L., Winker, D. M., Kar, J., Lee, K.-P., and Toth, T. D.: CALIPSO lidar calibration at 532 nm: version 4 daytime algorithm, Atmos. Meas. Tech., 11, 6309–6326, https://doi.org/10.5194/amt-11-6309-2018, 2018.

Hofer, J., Althausen, D., Abdullaev, S. F., Makhmudov, A. N., Nazarov, B. I., Schettler, G., Engelmann, R., Baars, H., Fomba, K. W., and Müller, K.: Long-term profiling of mineral dust and pollution aerosol with multiwavelength polarization Raman lidar at the Central Asian site of Dushanbe, Tajikistan: case studies, Atmos. Chem. Phys., 17, 14559-14577, https://doi.org/10.5194/acp-17-14559-2017, 2017.

Kim, M.-H., Omar, A. H., Tackett, J. L., Vaughan, M. A., Winker, D. M., Trepte, C. R., Hu, Y., Liu, Z., Poole, L. R., Pitts, M. C., Kar, J., and Magill, B. E.: The CALIPSO version 4 automated aerosol classification and lidar ratio selection algorithm, Atmos. Meas. Tech., 11, 6107–6135, https://doi.org/10.5194/amt-11-6107-2018, 2018.

Mamouri, R.-E., and Ansmann, A.: Potential of polarization/Raman lidar to separate fine dust, coarse dust, maritime, and anthropogenic aerosol profiles, Atmos. Meas. Tech., 10, 3403-3427, https://doi.org/10.5194/amt-10-3403-2017, 2017.

Müller, D., Wandinger, U., and Ansmann, A.: Microphysical particle parameters from extinction and backscatter lidar data by inversion with regularization: theory, Appl. Opt., 38, 2346-2357, https://doi.org/10.1364/AO.38.002346, 1999.

Schepanski, K., Tegen, I., and Macke, A.: Saharan dust transport and deposition towards the tropical northern Atlantic, Atmos. Chem. Phys., 9, 1173-1189, https://doi.org/10.5194/acp-9-1173-2009, 2009.

Vaughan, M., Garnier, A., Josset, D., Avery, M., Lee, K.-P., Liu, Z., Hunt, W., Pelon, J., Hu, Y., Burton, S., Hair, J., Tackett, J. L., Getzewich, B., Kar, J., and Rodier, S.: CALIPSO lidar calibration at 1064 nm: version 4 algorithm, Atmos. Meas. Tech., 12, 51–82, https://doi.org/10.5194/amt-12-51-2019, 2019.

- The inversion method requires information at three different wavelength. What is the procedure to match the CALIPSO and Aeolus profiles? (e.g. advection of CALIPSO profiles towards the Aeolus profile using ERA5 winds?)
As the time and space matching of the two observations cannot be perfect, how is the mismatch propagated into the error estimate?

AR: Thank you for your suggestion. As we explained above, since the orbits of Aeolus and CALIPSO are different, they cannot meet each other at the exactly same time and same location. In our study, the closest CALIPSO scanning tracks to those of Aeolus, are about 4 hours ahead of Aeolus. Based on the transport directions of dust events modelled with HYSPLIT, the tracks of Aeolus should be always downwind of the tracks of CALIPSO. When the tracks of Aeolus and CALIPSO are selected, the distances

between the tracks can be calculated. Assuming that the wind speed between CALIPSO scanning tracks and Aeolus scanning tracks is in the range of 5m·s$^{-1}$ to 15m·s$^{-1}$, the transport distances of the dust plumes are in the range of 72km to 216km. Besides, during the short-time transportation of Sahara dust plume, dust optical properties maintain almost unchanged (Haarig et al., 2017). Therefore, in our work, we think that during the short-time (~4 hours) transportation of Sahara dust plume, dust optical properties maintain almost unchanged. Consequently, in our study, if the distances between two satellites scanning tracks are less than 200 km and the tracks of Aeolus are downwind of the tracks of CALIPSO, it is reasonable to state that the dust plumes captured by CALIPSO are transported towards the Aeolus scanning regions in around 4 hours, hence the following procedures could be continued. From the measurement cases provided by the previous papers, the optical properties of Sahara dust within short-time (e.g., 2h, 4h…) seems to be stable. Besides, it has to be emphasized that, since we reject the data where meteorological data has more than 90 % of relative humidity, the changing of dust plumes in our study should be very slow without the effect of hygroscopicity. Hence, we insist that the combination of the optical properties provided by CALIPSO and Aeolus, with 4 hours difference, is applicable/reasonable to calculate the volume concentration and mass concentration.

To make it clear to the reader, we also add the similar explanations in the Section 3 of the revised manuscript. Please refer to the corresponding section. Thanks.

As for the calculation of the dust advection, the ERA5 wind field data between the tracks of Aeolus and CALIPSO at all height surfaces are smoothly distributed and the values are stable. It means that the Aeolus L2C data can be used at the location of the CALIPSO track.

*Reference:*

*Haarig, M., Ansmann, A., Althausen, D., Klepel, A., Groß, S., Freudenthaler, V., Toledano, C., Mamouri, R.-E., Farrell, D. A., Prescod, D. A., Marinou, E., Burton, S. P., Gasteiger, J., Engelmann, R., and Baars, H.: Triple-wavelength depolarization-ratio*

*profiling of Saharan dust over Barbados during SALTRACE in 2013 and 2014, Atmos.*
*Chem. Phys., 17, 10767–10794, https://doi.org/10.5194/acp-17-10767-2017, 2017.*

The combination of the two satellites observation is really interesting but the method really needs to be described precisely enough so that the results can be reproduced.

AR: Yes, we totally agree with you. In the revised manuscript, the detailed description of track-match method, datasets and quality control method, data processing and dust advection calculation are rewrite in the "Methodology". The improved method flowcharts are also updated in Figure 1 and Figure 2. All the data are re-produced accordingly.

l. 161: The authors reject data where meteorological data has more than 90 % of relative humidity. I suppose this is the method they chose for cloud screening but this needs to be stated explicitly. As said in the general comments, more information on data quality control and selection needs to be provided, e.g. Do you integrate all of the particles in a given cross-section? Could you discuss contamination by particles other than Saharan dust? (clouds, marine aerosols ...)

AR: Thanks. Aside for the cloud screening, when the relative humidity is larger than 90%, the dust aerosol will be influenced by the hygroscopicity effect and its properties could change (Engelmann et al., 2008). Then the mass concentration calculation method does not make sense any more. We state the corresponding explanation explicitly in the revised manuscript in Section 3 as below:

"Besides, when the RH is larger than 90%, the dust aerosol will be influenced by the hygroscopicity effect and its properties could change. Then the mass concentration calculation method does not make sense any more (Engelmann et al., 2008). For the cloud screening, aside the RH data, we use Level 2 5km aerosol profile of CALIPSO, which only provide aerosol optical properties so the cloud can be screened."

We do the dust advection calculation by every profile of each scanning track, not

by the integration of the given cross-sections.

For cloud screening, we use Level 2 5km aerosol profile of CALIPSO, which only provide aerosol optical properties so the cloud can be screened. We use Level 2 vertical feature mask of CALIPSO, which provide aerosol classification data, to get rid of the influence of other aerosol types. In other words, we only do the dust advection calculation when the aerosol type is "dust" or "polluted dust". Consequently, we insist that the contamination by particles other than Saharan dust can be negligible.

*Reference:*

*Engelmann, R., Wandinger, U., Ansmann, A., Müller, D., Žeromskis, E., Althausen, D., and Wehner, B.: Lidar observations of the vertical aerosol flux in the planetary boundary layer, J. Atmos. Ocean. Tech., 25, 1296-1306, https://doi.org/10.1175/2007JTECHA967.1, 2008.*

l. 163: The mass flux is said to be derived by eddy covariance, but it is not clear to me why we would consider turbulent transport and how this would be done. Is the mass flux derived as the integral of m.v for each pixel? Or is it " m'. v' " as stated? If it's actually the second option, how are m' and v' derived?

AR: Thank. Actually, we didn't implement mass flux calculation by eddy covariance because m' and v' can't be derived by Aeolus and CALIPSO profiles, sorry for the mistake. To quantificationally describe the dust transportation, the "dust advection" is defined as the multiplication of mass concentration (m) and wind velocity (v) of each data pixel in the revised manuscript. The corresponding definition and explanation are supplemented in section 3 "Methodology" of the revised manuscripts as below:

Ultimately, combining with the particle mass concentration and the horizontal wind speed provided by Aeolus and ECMWF, the dust mass advection is defined as Eq. (1), to represent the transportation of dust aerosol quantificationally.

$$\overrightarrow{Advection}_{\text{aerosol-mass}} = m \cdot \vec{v} \text{,} \tag{1}$$

where $m$ is the aerosol mass concentration and $v$ is the horizontal wind velocity.

As a second point, does this requires an interpolation of m and/or v on a common pixel grid? How is this done?

AR: Yes, the calculation of dust advection does require an interpolation of m and/or v on a common pixel grid. The procedure to acquire common pixel grid data is introduced here, as well in the Section 3 of the revised manuscripts. Firstly, vertical resolution and horizontal resolution between Aeolus data and CALIPSO data are different. For vertical resolution, "mid bin" optical property products of 23 data bins from Aeolus L2A are interpolated to 399 data bins from CALIPSO according to the altitude information of two products. For horizontal resolution, both Aeolus and CALIPSO products are averaged along every integer latitude to acquire a common horizontal pixel grid. For ECMWF wind field data, wind speed data, wind direction data and relative humidity data between Aeolus and CALIPSO scanning tracks are averaged along longitude and averaged along every integer latitude, while, vertically, it is interpolated to CALIPSO data bins to match the common pixel grid.

Fig. 4:
- Could you show the lidar profile from which the location of the stars is chosen?
AR: Thank you for your suggestion. We plot the CALIPSO total backscatter coefficient and particle depolarization ratio profiles separately of three chosen sites in the updated Fig.4 (also shown as below). From the separate profiles of these three sites, dust layers with high depolarization ratio can be recognized clearly. The existences of the dust layers are the pre-conditions of the HYSPLIT model simulation.

[Figure]

Figure 4. (a)(c)(e) CALIPSO total backscatter coefficient profiles and particle depolarization ratio profiles capturing dust layers at around 0500UTC on 20 June 2020. (b)(d)(f) HYSPLIT backward trajectories and forward trajectories ending at different sites of corresponding CALIPSO profiles and different heights on 0500UTC on 20 June 2020. The backward and forward trajectories' durations are both 144 hours.

- Using squares triangles and circles separately for each starting point would allow you to pack more information into this figure (e.g. northernmost point is associated with triangles, middle point with squares etc.).

AR: Yes, we agree with you. In the updated Fig.4, we implement HYSPLIT simulation separately to make the picture clearer to read. In the new figure, the CALIPSO aerosol profiles of three chosen positions along with the corresponding HYSPLIT backward trajectories and forward trajectories are provided separately, which may make the figure pack more information.

l.192: "deposition" is ambiguous. The HySPLIT data shows a downdraft of the air mass. The dust could also settle while the air mass has no vertical movement.

AR: Thanks. In the origin manuscripts, the statement "And an obvious deposition process of dust plumes is observed." may be ambiguous. We rephrased this statement as "At the end of these three forward trajectories, the altitudes of the dust aerosol reduce to around 1km, which indicates a downdraft process of dust plumes.". The decrease of the altitude in the end of HYSLPLIT forward trajectories and low altitude dust plumes observed by CALIPSO and Aeolus can be verified with each other. Therefore, we think that HYSPLIT indicates downdraft process of dust plumes is reasonable.

Fig. 5:
- Why show 4 lidar profiles but only exploit 3 of them? If it's not necessary, removing the first one would make the figure easier to read. Overall the 3D figures are pretty but contain too much information to be fully readable.

AR: Thanks. In Fig. 5 in the revised manuscript, we remove the first cross-section without dust to make the figure easier to read. Fig. 5 in the revised manuscript is also shown as below.

- On panel (a), could you label the date and time of each satellite overpass?

AR: Yes. Because the time of Aeolus and CALIPSO scanning tracks are different and the tracks are too close to each other on panel (a), the pass time of two satellites are

labeled separately on panel (b) and panel (c) in the revised manuscripts Fig. 5 which is shown below.

[Figure]

**2020-06-19 Aeolus and CALIPSO Descending Tracks**

**(a)**

lg(Exinction Coefficient, m⁻¹-ALADIN@355nm) 2020-06-19 Descending

**(b)**

lg(Total Backscatter Coefficient, m⁻¹sr⁻¹-CALIOP@532nm) 2020-06-19 Descending

**(c)**

l. 205-206: The origin of these numbers and of the associated error needs to be detailed

(cf comment on l. 160).

AR: These numbers are the mean backscatter coefficients at 532 nm of every cross-sections from CALIPSO. And the numbers behind"±" are the standard deviation, not the errors.

l. 225: same as l. 161, I suppose this is a method for rejecting cloud contaminated pixels. It would be better if this was stated explicitly.

AR: Thanks. As is mentioned above, aside for the cloud screening, when the relative humidity is larger than 90%, the dust aerosol will be influenced by the hygroscopicity effect and its properties could change. Then the mass concentration calculation method does not make sense any more (Engelmann et al., 2008). For cloud screening, we use Level 2 5km aerosol profile of CALIPSO, which only provide aerosol optical properties so the cloud can be screened. We state the corresponding explanation explicitly in the revised manuscript in Section 3.

*Reference:*
*Engelmann, R., Wandinger, U., Ansmann, A., Müller, D., Žeromskis, E., Althausen, D., and Wehner, B.: Lidar observations of the vertical aerosol flux in the planetary boundary layer, J. Atmos. Ocean. Tech., 25, 1296-1306, https://doi.org/10.1175/2007JTECHA967.1, 2008.*

l. 233-234: how can the dust concentration be highest in the cross-section 3, farthest from the source. I would expect dispersion of aerosols rather than concentration. Could this be justified?

AR: In the revised manuscript, thanks to your recommendations/suggestions in the usage of "Mid_bin" of L2A data, we re-produced the calculation of the mass

concentration and dust advection. It is figured out that the mean dust advection value is about $2.13\,\mathrm{mg\cdot m^{-2}\cdot s^{-1}}$ in "cross-section 1" (dust portion during emission phase), $1.55\,\mathrm{mg\cdot m^{-2}\cdot s^{-1}}$ in "cross-section 2" (dust portion during development phase) and $0.51\,\mathrm{mg\cdot m^{-2}\cdot s^{-1}}$ in "cross-section 3" (dust portion during deposition phase). Hence actually the mean advection value in the cross-section 3 is the lowest. Perhaps it because that the cross-section 3 is farthest from the source region. During the dispersion and deposition processes of dust aerosol transportation, it is reasonable that the lowest value appears in cross-section 3. Fig. 7 in the revised manuscript is also shown below:

[Figure]

Fig. 7 and Fig. 10: Estimations of mass flux are provided with an error bar but it is nowhere described how these error bars are derived. We need some proper description

of this.

AR: Thank you for your suggestion. The error bars in these figures (in original manuscript) are not the estimated errors, but the standard deviations of every cross-section. In the revised manuscript, since we have discussed the error estimations in calculating volume concentration and mass concentration, we omit these bars since the statistical deviations seem to be meaningless here. Fig. 10 in the revised manuscript is also shown below:

[Figure]

Fig. 8: There is too much on this figure. It is hard to read. You need to simplify the presentation.

AR: Thank you for your suggestion. We simplify and beautify Fig. 8 in the revised manuscripts. The HYSPLIT trajectories curves of the whole figure are bolded and the backscatter coefficient cross-sections at 532 nm from CALIPSO are removed. Fig. 8 in the revised manuscript is shown as below:

[Figure]

l. 276-284: It is no obvious from Fig. 11 that the low chlorophyll concentration is due

to a lack of iron. More references would be needed to support this claim.

AR: Thank you for your suggestion. Relevant statements and references have been supplemented in the revised manuscript and they are demonstrated below.

According to Mills et al. (2004), in the tropical North Atlantic, community primary productivity was nitrogen-limited, and that nitrogen fixation was co-limited by iron and phosphorus. Saharan dust addition stimulated nitrogen fixation, presumably by supplying both iron and phosphorus. The mineral dust contains micronutrients such as Fe and P that have the potential to act as a fertilizer, increasing primary productivity in the equatorial Atlantic Ocean, and thus leading to $N_2$ fixation and $CO_2$ drawdown (Meskhidze et al., 2007). Consequently, the dust plumes observed in this study could be the fertilizer of Atlantic Ocean and the influence of the dust plumes deposition will be in our future research.

*Reference:*

*Meskhidze, N., Nenes, A., Chameides, W. L., Luo, C., & Mahowald, N.: Atlantic Southern Ocean productivity: Fertilization from above or below?. Global Biogeochemical Cycles, 21(2). https://doi.org/10.1029/2006GB002711, 2007.*

*Mills, M. M., Ridame, C., Davey, M., La Roche, J., & Geider, R. J.: Iron and phosphorus co-limit nitrogen fixation in the eastern tropical North Atlantic. Nature, 429(6989), 292-294. https://doi.org/10.1038/nature02550, 2004.*

Conclusion: The L2A data was not available publicly at the time the article was submitted. Authors could state that they were allowed to access the data through their participation as a Calibration and Validation team.

AR: Yes. We added this statement in the "data availability" section. Thanks for reminding.

---

## Author Comment (AC2)

Response to Reviewer CC:

Section 2.1 ALADIN/Aeolus:

Indicate which L2A baseline (i.e. baseline 10 or 11 referring to the L2Ap v3.10 or v3.11) has been used for processing might be useful for traceability (e.g. new radiometric correction being included in v3.11 using telescope temperatures oscillations).

AR: Thank you for your suggestion. For L2A products and L2C products from Aeolus, we both use baseline 10 data (referring to the L2Ap v3.10). In the revised manuscript, the corresponding explanation is supplemented as following:

"In this study, the Level 2A (baseline 10 referring to the L2Ap v3.10) aerosol optical properties and Level 2C (baseline 10 referring to the L2Ap v3.10) wind vectors are used."

Section 4.1 Measurement case with CALIOP, ALADIN, ECMWF and HYSPLIT:

The differences between Aeolus/ALADIN and CALIPSO/CALIOP instrumentation principle and geometry could be highlighted (i.e. ALADIN poiting 35° offset from nadir with the ground) as the time gap between acquisitions (e.g. for intercompaison cross-section 3 on June 19, 2020 showed in Figure 5 Aeolus hovered the West of Cape Verde from 08:00 to 08:30 UTC four hours later than CALIOP from 04:07 to 04/20 UTC). It is fair to assume that the particles distribution within the plume might have evolved during the time offset, hence a limit of the data intercomparison.

AR: Thanks. Since the instrumentation principle and geometry of Aeolus/ALADIN and CALIPSO/CALIOP is different and the orbits of Aeolus/ALADIN and CALIPSO/CALIOP are not coincide, there are always space gaps and time gaps (around 4 hours) between the closest two scanning tracks of Aeolus and CALIPSO. Although the existence of the space gaps and time gaps of two satellites scanning tracks, it is considered that the optical properties of the dust plumes between two satellites

scanning tracks are almost unchanged.

In our study, the closest CALIPSO scanning tracks to those of Aeolus, are about 4 hours ahead of Aeolus. Based on the transport directions of dust events modelled with HYSPLIT, the tracks of Aeolus should be always downwind of the tracks of CALIPSO. When the tracks of Aeolus and CALIPSO are selected, the distances between the tracks can be calculated. Assuming that the wind speed between CALIPSO scanning tracks and Aeolus scanning tracks is in the range of 5m·s⁻¹ to 15m·s⁻¹, the corresponding transport distances of the dust plumes are in the range of 72km to 216km.

[Figure]

Figure 1. Saharan air layer (from 1.2 to 4.2 km height) above the marine boundary layer. Panel (b) shows the range-corrected cross-polarized backscatter signal at 532 nm with temporal resolution of 30 s and vertical resolution of 7.5 m, respectively. Panel (d) shows the volume linear depolarization ratio at 532 nm. The radiosonde profiles of wind speed (WS) and wind direction (WD) are shown in panel (a), the profiles of relative humidity (RH) and temperature (T) in panel (c). The lidar observation was performed on 6 July 2014, 23:18–01:33 UTC. The radiosonde was launched at

23:47 UTC (indicated by a black vertical line). [Figure 11 from Haarig et al. (2017).]

Figure 1 shows the wind direction, wind speed, relative humidity and temperature measured by radiosonde, as well as the range-corrected cross-polarized Sahara dust backscatter signal at 532 nm and the Sahara dust volume linear depolarization ratio at 532 nm measured by BERTHA (Backscatter Extinction lidar-Ratio Temperature Humidity profiling Apparatus) at Barbados, which is located in Eastern Caribbean (Haarig et al., 2017). It can be seen from Figure 1 that even after long-range transportation, the optical properties (volume linear depolarization ratio at 532 nm) of the dust layer maintain stable in about 2 hours. Therefore, in our work, we think that during the short-time (~4 hours) transportation of Sahara dust plume, dust optical properties maintain almost unchanged.

Consequently, in our study, if the distances between two satellites scanning tracks are less than 200 km and the tracks of Aeolus are downwind of the tracks of CALIPSO, it is reasonable to state that the dust plumes captured by CALIPSO are transported towards the Aeolus scanning regions in around 4 hours, hence the following procedures could be continued. From the measurement cases provided by the paper mentioned above, the optical properties of Sahara dust within short-time (e.g., 2h, 4h…) seems to be stable. Hence, we insist that the combination of the optical properties provided by CALIPSO and Aeolus, with 4 hours difference, is applicable/reasonable to calculate the volume concentration and mass concentration.

To make it clear to the reader, we also add the similar explanations in the Section 3 of the revised manuscript as following:

"Assuming the wind speed between CALIPSO scanning tracks and Aeolus scanning tracks is in the range of 5 $m \cdot s^{-1}$ to 15 $m \cdot s^{-1}$, the transport distances of the dust plumes are in the range of 72km to 216km. Besides, during the short-time transportation of Sahara dust plume, dust optical properties maintain almost unchanged (Haarig et al., 2017). Consequently, in our study, if the distances between two satellites scanning tracks are less than 200 km and the tracks of Aeolus are downwind of the tracks of CALIPSO, it is reasonable to state that the dust plumes captured by CALIPSO are transported towards the Aeolus scanning regions in around 4 hours, hence the

following procedures could be continued."

*Reference:*

*Haarig, M., Ansmann, A., Althausen, D., Klepel, A., Groß, S., Freudenthaler, V., Toledano, C., Mamouri, R.-E., Farrell, D. A., Prescod, D. A., Marinou, E., Burton, S. P., Gasteiger, J., Engelmann, R., and Baars, H.: Triple-wavelength depolarization-ratio profiling of Saharan dust over Barbados during SALTRACE in 2013 and 2014, Atmos. Chem. Phys., 17, 10767–10794, https://doi.org/10.5194/acp-17-10767-2017, 2017.*

---

## Author Comment (AC3)

Response to Reviewer #2:

In the present manuscript, Guangyao Dai and coauthors track a Saharan dust plume across the Atlantic Ocean and calculate the dust horizontal fluxes. The novel approach in their manuscript is the combination of two satellites (CALIPSO and Aeolus) measuring at different wavelengths. To bridge the gap between the overpasses of the two satellites ERA5 model reanalysis and HYSPLIT trajectories are used. However, the satellite data are not treated in a correct manner with the result that the whole proposed method is not valid. Therefore, I have to reject the manuscript.

AR: Thanks. The train of thought of this work is using CALIPSO and Aeolus aerosol optical properties to capture and describe a long-range Sahara dust transportation event which occurred from 15 June 2020 to 27 June 2020. ERA5 model reanalysis wind field data and HYSPLIT trajectories are used as tools to verify the whole transportation. Finally, dust mass concentration derived from **five** aerosol optical properties, which are **backscatter coefficients, extinction coefficients at 532nm,1064nm from CALIPSO** and **extinction coefficient at 355nm** from Aeolus, combined with ERA5 relative humidity, wind field data which assimilate Aeolus L2B HLOS wind data to implement calculation of dust advection, which is defined as the multiplication of dust mass concentration and horizontal wind velocity.

However, it is surely a challenge of this work **based on present technology** because the time and distance gaps of CALIPSO and Aeolus overpasses, and the developing Aeolus L2A product which needs new algorithm (Flament et al., 2021). More improvements and efforts need to be implemented to acquire more precise analysis and calculation. Nevertheless, in this work, based on present technology, we utilize present data and model as carefully as possible. Therefore, we think this work is acceptable and reasonable.

Hence, after the comprehensive consideration, the treatment for the satellites' data and the proposed method are improved and updated in section 3 "Methodology" of the revised manuscript. The corresponding corrections are presented as following:
* * *
**3. Methodology**

In the study of dust transport and advection measurement, as shown in Figure 1, the dust identification, Aeolus and CALIPSO tracks match, data analysis and the HYSPLIT model analysis are necessary and the schematic flowchart is described briefly.

**3.1 Method used to match CALIPSO and Aeolus data**

To identify the dust events and to choose the quasi-synchronization observations with ALADIN and CALIOP, the flowchart is presented in this figure. To preliminarily determine the occurrences of dust events, the "Dust score index" data provided by AIRS/Aqua are used to determine the dust plume coverage and transport route. With this given information, the VFM products from the simultaneously observations with the spaceborne lidar CALIOP are applied to cross-check the identification of dust events. Hence the vertical distributions of dust plumes are presented. To find the original sources and to predict the transport routes of dust plumes, the backward trajectory and forward trajectory is used respectively. When the dust events are determined, the simultaneous observations with ALADIN and CALIOP have to be selected. As the dust plumes can be captured by CALIPSO VFM products, hence, starting from the CALIOP observations, the nearest Aeolus footprints could be figured out. Since the orbits of Aeolus and CALIPSO are different, they cannot meet each other at the exactly same time and same location. From our study, the closest CALIPSO scanning tracks to those of Aeolus, are about 4 hours ahead of Aeolus. Based on the transport directions of dust events modelled with HYSPLIT, the tracks of Aeolus should be always downwind of the tracks of CALIPSO. When the tracks of Aeolus and CALIPSO are selected, the distances between the tracks can be calculated. Assuming the wind speed scale between CALIPSO scanning tracks and Aeolus scanning tracks is $5 \ \mathrm{m \cdot s^{-1}}$ to $15 \ \mathrm{m \cdot s^{-1}}$, the transport distance scale of the dust plumes is 72km to 216km. Besides, during the short-time transportation of Sahara dust plume, dust optical properties maintain almost unchanged (Haarig et al., 2017). Consequently, in our study, if the distances between two satellites scanning tracks are less than 200 km and the tracks of Aeolus are downwind of the tracks of CALIPSO, it is reasonable to state that the dust plumes captured by CALIPSO are transported towards the Aeolus scanning regions in around 4 hours, hence the following procedures could be continued.

**3.2 Datasets and quality control**

In this study, the extinction coefficient at 355 nm from ALADIN, at 532 nm and 1064 nm from CALIOP are collected as the useful dataset. The extinction coefficients at 355 nm correspond to the "Aeolus Level 2A Product" retrieved by SCA (standard correct algorithm). In this study, we choose SCA instead of ICA (iterative correct algorithm) because the extinction coefficients from ICA are noisy and the assumption of "one

single particle layer filling the entire range bin" in SCA is reasonable and is met in the situation of the heavy dust events observation. Additionally, we use the mid bin product (sca_optical_properties_mid_bins) of SCA instead of the normal product of SCA, because the mid-bin algorithm provides more robust results (Baars et al., 2021; Flament et al., 2021). The extinction coefficients, which are more sensitive to noise and are the significant inputs of the dust advection calculation, are better retrieved through this "mid bin" averaged version of the algorithm. In terms of quality control, negative extinction coefficient values of L2A are excluded while the "bin_1_clear" flag and the "processing_qc_flag" of L2A are used to eliminate invalid data. The backscatter coefficients and extinction coefficients at 532 nm and 1064 nm are the "Total_Backscatter_Coefficient_532", "Extinction_Coefficient_532", "Backscatter_Coefficient_1064" and "Extinction_Coefficient_1064". Moreover, "Extinction_QC_Flag_532" and "Extinction_QC_Flag_532" from CALIPSO Level 2 products are used to conduct quality control of CALIPSO data. Since the footprints of Aeolus and CALIPSO are not exactly matched, the missing wind data between their tracks have to be filled in using the ERA5 wind field data. There are two purposes on the usage the ERA5 wind field data between Aeolus and CALIPSO tracks. One is that the ERA5 wind speed and direction data provides the evidence of dust transporting from CALIPSO tracks towards Aeolus tracks. Besides, the ERA5 wind field data between the tracks of Aeolus and CALIPSO at all height surfaces are smoothly distributed and the values are stable. It means that the Aeolus L2C data can be used at the location of the CALIPSO track.

**3.3 Dust advection calculation**

In Figure 2, the flowchart of dust mass advection calculation procedure is provided. Based on the dataset consists of the backscatter coefficients and extinction coefficients at the wavelengths of 1064 nm and 532 nm from CALIOP and the extinction coefficients at the wavelength of 355 nm from ALADIN, the aerosol volume concentration distribution can be calculated based on regularization method which was performed by generalized cross-validation (GCV) from Müller et al. (1999).

The advantage of this method is that it does not require prior knowledge of the shape of the particle size distribution and the estimate uncertainty of aerosol volume concentration is on the order of 50% if the estimated errors of the input is on the order of 20%. For the accuracy of the CALIPSO-retrieved extinction and backscatter coefficients: for the backscatter coefficient at 532 nm, during the daytime, the average difference between collocated CALIPSO and HSRL measurements is 1.0%±3.5 % in V4 (Getzewich et al., 2018); for the backscatter coefficient at 1064 nm, the CALIOP V4 1064 nm calibration coefficients are accurate to within 3 % (Vaughan et al., 2019); for the extinction coefficients, the uncertainty in the V4 dust lidar ratio of 20 % (30 %) at 532 nm (1064 nm) (Kim et al., 2018), thus it is considered that the estimate errors of the extinction coefficients from CALIPSO are on the order of 20%. Consequently, we think that the uncertainties of CALIPSO-retrieved extinction and backscatter coefficients are on the order of 20%. According to Flament et al. (2021), because of the lack of cross-polarized light, 355nm backscatter coefficients of Aeolus are

underestimated, especially for dust aerosol. Nevertheless, the extinction is not affected. In this work, Aeolus-retrieved backscatter coefficients at 355nm are not applied for the calculations of the dust volume concentration distribution and mass concentration. For the accuracy of the Aeolus-retrieved extinction coefficient, the simulation extinction coefficients fit the inputs well mostly, especially when the altitude is larger than 2km (Flament et al, 2021). Hence, we think that after rigorous quality control, Aeolus L2A extinction coefficient could be the input parameters of the regularization method. In conclusion, we think that the estimated errors of the five input parameters we used to calculate the aerosol volume concentration are on the order of 20%. The estimate errors of dust advection are the combination of mass concentration estimate errors (~50%) and Aeolus L2C wind vector estimate errors.

It should be emphasized that due to the different vertical resolution and horizontal resolution between Aeolus data and CALIPSO data, a common pixel grid should be conducted before calculation. For vertical resolution, 23 data bins of Aeolus L2A mid bin optical property products are interpolated to 399 data bins of CALIPSO according to the altitude information of two products. For horizontal resolution, both Aeolus and CALIPSO products are averaged along every integer latitude to acquire a common horizontal pixel grid. After integrating and multiplying an assuming typical dust particle density which is set as 2.65 $\mathrm{g \cdot cm^{-3}}$ referring to previous studies (e.g., Schepanski et al., 2009; Hofer et al., 2017; Mamouri and Ansmann, 2017), the particle mass concentration would be estimated as Engelmann et al. (2008) introduced. For ECMWF wind field data, wind speed data, wind direction data and RH data between Aeolus and CALIPSO scanning tracks are averaged along longitude and averaged along every integer latitude, while, vertically, they are interpolated to CALIPSO data bins to match the common pixel grid. Besides, when the RH is larger than 90%, the dust aerosol will be influenced by the hygroscopicity effect and its properties could change. Then the mass concentration calculation method does not make sense any more (Engelmann et al., 2008). For the cloud screening, aside the RH data, we use Level 2 5km aerosol profile of CALIPSO, which only provide aerosol optical properties so the cloud can be screened. Therefore, relative humidity data provided by ECMWF is used to filtrate unavailable data. Ultimately, combining with the particle mass concentration and the horizontal wind speed provided by Aeolus and ECMWF, the dust mass advection is defined as Eq. (1), to represent the transportation of dust aerosol quantificationally.

$$\overrightarrow{Advection}_{\mathrm{aerosol-mass}} = m \cdot \vec{v}, \tag{1}$$

where $m$ is the aerosol mass concentration and $\vec{v}$ is the horizontal wind velocity.

[Figure]

**Figure 1. Dust identification, Aeolus and CALIPSO tracks match and data procedures.**

[Figure]

**Figure 2. The flowchart of the dust mass advection calculation procedure.**
* * *
*Reference:*

*Flament, T., Trapon, D., Lacour, A., Dabas, A., Ehlers, F., and Huber, D.: Aeolus L2A Aerosol Optical Properties Product: Standard Correct Algorithm and Mie Correct Algorithm, Atmos. Meas. Tech. Discuss. [preprint], https://doi.org/10.5194/amt-2021-181, in review, 2021.*

The following points underline my decision and may help the authors to improve their work:

Aeolus is providing the circular co-polarized component of the backscatter and not the total backscatter coefficient. The missing cross-polarized component is not negligible

in dust cases as used in the manuscript. You are missing a significant part of the backscatter coefficient at 355 nm.

AR: Thank you for your suggestion.

As reported in Flament et al. (2021), it is clearly stated that "Designed as a wind lidar, ALADIN does not have the ability to measure depolarization. The UV laser beam is linearly polarized, and analyzed only along the parallel direction. Any cross-polarized light is rejected. This means that, in order to compare Aeolus observations to other instruments, only the co-polar component must be considered. Not going through this extra step before comparing would make it seem that the total backscatter of highly depolarizing targets such as ice crystals or dust is largely underestimated by Aeolus. **Because the extinction is not affected,** the corresponding Aeolus lidar ratio is going to be larger than the total lidar ratio.". It can be concluded that because of the missing cross-polarized component, Aeolus backscatter coefficient at 355nm is underestimated, especially for dust aerosol. **Nevertheless, it has to be emphasized that the extinction coefficient at 355nm (which is used in our research) is not affected.** Hence, in our study, **we applied Aeolus-retrieved extinction coefficient instead of backscatter coefficient at 355nm during the calculation of volume concentration and mass concentration.**

As described in the section 3 of the revised manuscript, we only use five aerosol optical properties to estimate dust volume concentration and mass concentration, which are backscatter coefficients and extinction coefficients at 532nm and 1064nm from CALIPSO and extinction coefficient at 355nm from Aeolus. In our study, Aeolus-retrieved backscatter coefficients at 355nm are not applied for the calculations of the dust volume concentration distribution and mass concentration. Thus, we insist that the usage of Aeolus-retrieved extinction coefficient in calculating the dust volume concentration and mass concentration is reasonable.

Besides, Aeolus can provide valuable information thanks to its HSRL design. Flament et al. (2021) also proves that Aeolus has the ability to capture dust aerosol layers, which are from the same dust transportation event as this work (as shown below).

[Figure]

*Reference:*

*Flament, T., Trapon, D., Lacour, A., Dabas, A., Ehlers, F., and Huber, D.: Aeolus L2A Aerosol Optical Properties Product: Standard Correct Algorithm and Mie Correct Algorithm, Atmos. Meas. Tech. Discuss. [preprint], https://doi.org/10.5194/amt-2021-181, in review, 2021.*

CALIPSO measures the backscatter coefficient at 532 and 1064 nm, but not the extinction. The extinction provided by CALIPSO is retrieved by multiplying the backscatter coefficient with the aerosol-type-dependent lidar ratio. Therefore, the extinction coefficient is not an independent quantity. For your inversion calculation, you need independent measurements of the extinction coefficient, either by high spectral resolution (HSRL) or Raman lidar measurements.

AR: Thank you for your suggestion. Because of the detection principle, CALIPSO can only derive 532nm and 1064nm backscatter coefficient directly. The extinction retrieval of CALIPSO definitely needs more complex and aerosol-type-dependent algorithms. After the launch of CALIPSO, the retrieval algorithm of extinction had been established and developed, which was named as Hybrid Extinction Retrieval Algorithms (HERA) (Young et al., 2009). Uncertainty and error sensitivity analyses of this HERA algorithm were implemented to evaluate the propagation of uncertainty errors and bias errors (Young et al., 2013). To further evaluate the errors of extinction and improve the extinction products quality, a large amount of validation campaigns and experiments

are implemented (Kacenelenbogen et al., 2011; Misra et al., 2012; Mioche et al., 2010). Recently, Abdoul et al. (2020) use the CALIPSO extinction observations to assess the performance of dust extinction coefficients modeled by the Weather Research and Forecasting model with Chemistry (WRF-Chem). Besides, Xing et al. (2021) combine aerosol extinction vertical profiles from the CALIPSO and assimilated multi-layer wind profiles from the MERRA-2 to calculate aerosol extinction flux.

Meanwhile, as reported in Getzewich et al. (2018): "Extensive validation data acquired by NASA's airborne high spectral resolution lidar (HSRL) shows that during the daytime the average difference between collocated CALIPSO and HSRL measurements of 532 nm attenuated backscatter coefficients is reduced from 3.3%±3.1 % in V3 to 1.0%±3.5 % in V4.". In Vaughan et al. (2019): "By evaluating calibration coefficients derived using both water clouds and ocean surfaces as alternate calibration targets, and through comparisons to independent, collocated measurements made by airborne high spectral resolution lidar, we conclude that the CALIOP V4 1064 nm calibration coefficients are accurate to within 3 %.". And in Kim et al. (2018): "The uncertainty in the V4 dust lidar ratio of 20 % (30 %) at 532 nm (1064 nm) accounts for the regional variability.". Therefore, we think that although the CALIPSO extinction coefficients are not independent quantities, but thanks to the numerous validation campaigns and the algorithm update, the CALIOP-retrieved extinction coefficients can be the credible parameters in the mass concentration calculation.

Besides, since the observation objects in our study is Sahara dust plumes, the dust lidar ratios are well-studied, e.g., Ansmann et al., 2011; Haarig et al., 2017 and the citations in these papers. With the CALIOP-retrieved backscatter coefficients, combining the Sahara dust lidar ratios, the Sahara dust extinction coefficients should be trustable.

*Reference:*

*Ansmann, A., Petzold, A., Kandler, K., Tegen, I. N. A., Wendisch, M., Mueller, D., Weinzierl, B., Mueller, T., and Heintzenberg, J.: Saharan Mineral Dust Experiments SAMUM–1 and SAMUM–2: what have we learned?. Tellus B: Chemical and Physical Meteorology, 63(4), 403-429. https://doi.org/10.1111/j.1600-0889.2011.00555.x, 2011.*

Chaibou, A. A. S., Ma, X., Kumar, K. R., Jia, H., Tang, Y., & Sha, T.: Evaluation of dust extinction and vertical profiles simulated by WRF-Chem with CALIPSO and AERONET over North Africa. Journal of Atmospheric and Solar-Terrestrial Physics, 199, 105213. https://doi.org/10.1016/j.jastp.2020.105213, 2020.

Haarig, M., Ansmann, A., Althausen, D., Klepel, A., Groß, S., Freudenthaler, V., Toledano, C., Mamouri, R.-E., Farrell, D. A., and Prescod, D. A.: Triple-wavelength depolarization-ratio profiling of Saharan dust over Barbados during SALTRACE in 2013 and 2014, Atmos. Chem. Phys., 17, 10767-10794, https://doi.org/10.5194/acp-17-10767-2017, 2017.

Kacenelenbogen, M., Vaughan, M. A., Redemann, J., Hoff, R. M., Rogers, R. R., Ferrare, R. A., Russell, P. B., Hostetler, C. A., Hair, J. W., and Holben, B. N.: An accuracy assessment of the CALIOP/CALIPSO version 2/version 3 daytime aerosol extinction product based on a detailed multi-sensor, multi-platform case study, Atmos. Chem. Phys., 11, 3981–4000, https://doi.org/10.5194/acp-11-3981-2011, 2011.

Mioche, G., Josset, D., Gayet, J. F., Pelon, J., Garnier, A., Minikin, A., & Schwarzenboeck, A.: Validation of the CALIPSO-CALIOP extinction coefficients from in situ observations in midlatitude cirrus clouds during the CIRCLE-2 experiment. Journal of Geophysical Research: Atmospheres, 115(D4). https://doi.org/10.1029/2009JD012376, 2010.

Misra, A., Tripathi, S. N., Kaul, D. S., & Welton, E. J.: Study of MPLNET-derived aerosol climatology over Kanpur, India, and validation of CALIPSO level 2 version 3 backscatter and extinction products. Journal of Atmospheric and Oceanic Technology, 29(9), 1285-1294. https://doi.org/10.1175/JTECH-D-11-00162.1, 2012.

Xing, Z., Li, S., Xiong, Y., & Du, K.: Estimation of cross-boundary aerosol flux over the Edmonton-Calgary Corridor in Canada based on CALIPSO and MERRA-2 data during 2011–2017. Atmospheric Environment, 246, 118084. https://doi.org/10.1016/j.atmosenv.2020.118084, 2021.

Young, S. A., & Vaughan, M. A.: The retrieval of profiles of particulate extinction from Cloud-Aerosol Lidar Infrared Pathfinder Satellite Observations (CALIPSO) data: Algorithm description. Journal of Atmospheric and Oceanic Technology, 26(6), 1105-

*1119. https://doi.org/10.1175/2008JTECHA1221.1, 2009.*

*Young, S. A., Vaughan, M. A., Kuehn, R. E., & Winker, D. M.: The retrieval of profiles of particulate extinction from Cloud–Aerosol Lidar and Infrared Pathfinder Satellite Observations (CALIPSO) data: Uncertainty and error sensitivity analyses. Journal of Atmospheric and Oceanic Technology, 30(3), 395-428. https://doi.org/10.1175/JTECH-D-12-00046.1, 2013.*

Following point 1 and 2, the main part of your data procedure, the calculation of the dust volume concentration is not correct. It can not be done in the presented manner. This is not an easy point to correct and leads to my decision to reject the paper.

AR: As replied above, for the point 1 and 2, we have explained/addressed them in detail. We insist that the datasets from ALADIN/Aeolus and CALIOP/CALIPSO and the updated methodology in the revised manuscript should be solid. We kindly ask for your reconsideration. Thanks.

The horizontal flux is not well defined. The horizontal velocity is a vector with two components (East-West, North-South), so the horizontal flux should have a direction. If you just take the absolute value of the velocity, your flux may have different directions at every point. What does this help us in understanding the dust transport?

AR: Yes, thanks for your suggestion. Firstly, it should be emphasized that, in the revised manuscript, we define "dust advection" instead of "mass flux" to describe dust transportation quantificationally. The "dust advection" is the multiplication of the mass concentration (m) and the horizontal wind velocity (v), which means it is a vector. In Fig. 7 and Fig. 10 of the revised manuscript, we plot the dust advection directions of every cross-section on panel (b) to explain the dust transport. Fig. 7 and Fig. 10 are shown below. It can be seen from panel (b)s of Fig. 7 and Fig. 10 that the dust advection directions at every point of every cross-section are shown clearly.

[Figure]

Fig. 7 The dust advection calculated with data from ALADIN, CALIOP and ECMWF

(a) the dust advection values at different cross-sections of dust plumes and (b) the dust

advection directions at different cross-sections of dust plumes on 19 June 2020.

[Figure]

Fig. 10 The dust advection calculated with data from ALADIN, CALIOP and ECMWF (a) dust advection values at different cross-sections and at different times during the dust transport and (b) dust advection directions at different cross-sections and at different times during the dust transport

Your result, that the minimum flux occurs at dust emission (line 271 and 322) is misleading. Why should the flux be lowest at emission? Looking at your back trajectories (Fig. 8a) indicates that a significant amount of dust originated from regions west of the track on 15 June. This dust is not observed on 15 June, but on 16 June leading to a greater horizontal flux.

AR: Yes, we agree with you. thanks to Thomas Flament's recommendations/suggestions in the usage of "Mid_bin" of L2A data, we re-produced the calculation of the mass concentration and dust advection. It is figured out by the revised calculation that the mean dust mass advection are about $1.67 \, \mathrm{mg \cdot m^{-2} \cdot s^{-1}}$ on 15 June 2020, $1.88 \, \mathrm{mg \cdot m^{-2} \cdot s^{-1}}$ on 16 June 2020, $1.55 \, \mathrm{mg \cdot m^{-2} \cdot s^{-1}}$ on 19 June 2020, $0.78 \, \mathrm{mg \cdot m^{-2} \cdot s^{-1}}$ on 24 June 2020 and $0.38 \, \mathrm{mg \cdot m^{-2} \cdot s^{-1}}$ on 27 June 2020. Actually, the mean dust advection value on 15 June is not the lowest anymore, but indeed lower than that on 16 June. In case of misleading, we addressed this statement in the revised manuscript. The slightly lower mean dust advection value occurs at dust emission on 15 June than that on 16 June results from the fact that the dust plume captured by CALIPSO and Aeolus is not the entire sources of the whole dust transportation. In the revised manuscript, it is explained as "It has to be emphasized that, according to Fig. 8(a), Aeolus and CALIPSO quasi-synchronically observed the dust plumes only at part (not whole) of the emission regions. The emission part from the West Africa (perhaps stronger than that from the central Africa) is missed and thus leading to the lower mean dust advection value on 15 June than that on 16 June."

The combination of the two satellites is a great new idea. However, you should highlight

the scientific question behind. You speak about ocean fertilization, but it remains open, which amount of dust is deposited to the Ocean. With Fig. 11, you show the low chlorophyll concentration in the studied area, but you do not quantify the effect of the discussed dust event on the ocean fertility. Your description remains very general stating that dust add nutrients to the Ocean.

AR: Yes. We only observe the long-range Sahara dust transportation by Aeolus, CALIPSO and reanalysis data and attempt to describe this event quantificationally. Actually, the study focusing on the ocean fertilization affected by Sahara dust is part of our ongoing work. According to Mills et al. (2004), in the tropical North Atlantic, community primary productivity was nitrogen-limited, and that nitrogen fixation was co-limited by iron and phosphorus. Saharan dust addition stimulated nitrogen fixation, presumably by supplying both iron and phosphorus. The mineral dust contains micronutrients such as Fe and P that have the potential to act as a fertilizer, increasing primary productivity in the equatorial Atlantic Ocean, and thus leading to $N_2$ fixation and $CO_2$ drawdown (Meskhidze et al., 2007). Consequently, the dust plumes observed in this study could be the fertilizer of Atlantic Ocean and the influence of the dust plumes deposition will be considered in our future research.

*Reference:*

*Meskhidze, N., Nenes, A., Chameides, W. L., Luo, C., & Mahowald, N.: Atlantic Southern Ocean productivity: Fertilization from above or below?. Global Biogeochemical Cycles, 21(2). https://doi.org/10.1029/2006GB002711, 2007.*

*Mills, M. M., Ridame, C., Davey, M., La Roche, J., & Geider, R. J.: Iron and phosphorus co-limit nitrogen fixation in the eastern tropical North Atlantic. Nature, 429(6989), 292-294. https://doi.org/10.1038/nature02550, 2004.*